# Multi-Class Support Vector Machine with Differential Privacy

**Jinseong Park[1],    Yujin Choi[2],    Jaewook Lee[2]\***

[1]Korea Institute for Advanced Study        [2]Seoul National University

jinseong@kias.re.kr, {uznhigh, jaewook}@snu.ac.kr

## Abstract

With the increasing need to safeguard data privacy in machine learning models, differential privacy (DP) is one of the major frameworks to build privacy-preserving models. Support Vector Machines (SVMs) are widely used traditional machine learning models due to their robust margin guarantees and strong empirical performance in binary classification. However, applying DP to multi-class SVMs is inadequate, as the standard one-versus-rest (OvR) and one-versus-one (OvO) approaches repeatedly query each data sample when building multiple binary classifiers, thus consuming the privacy budget proportionally to the number of classes. To overcome this limitation, we explore all-in-one SVM approaches for DP, which access each data sample only once to construct multi-class SVM boundaries with margin maximization properties. We propose a novel differentially Private Multi-class SVM (PMSVM) with weight and gradient perturbation methods, providing rigorous sensitivity and convergence analyses to ensure DP in all-in-one SVMs. Empirical results demonstrate that our approach surpasses existing DP-SVM methods in multi-class scenarios.

## 1 Introduction

As machine learning models may contain sensitive information about training data samples, privacy-preserving machine learning methods are actively investigated. Differential privacy (DP) [1, 2] is one of the prominent privacy concepts by offering a rigorous mathematical framework to quantify and bound the risk of disclosing a single individual's data in training datasets. To hide personal information, DP methods add random perturbations to model parameters or their outputs [3]. At the same time, as the randomness inevitably degrades the utility of the models, it is important to reduce the noise level or the number of data accesses [4].

Support vector machine (SVM) [5] is one of the widely used traditional machine learning models with a strong theoretical guarantee of margin and following empirical performance in binary classification tasks. Within various privacy-preserving SVMs [6, 7], Chaudhuri et al. [3] proposed a DP convex optimization approach and applied it to SVM with convex margin maximization to ensure DP within the SVM framework. Later research has focused on improving the convex optimization analysis to reduce noise levels [8–10]. Alternatively, previous papers tailored to DP-SVM frameworks [11, 12] have proposed enhancing SVM privacy using a Wolfe dual formulation. However, the multi-class classification using DP-SVMs has not been actively investigated. In multi-class classification, Park et al. [13] argued that traditional one-vs-rest (OvR) or one-vs-one (OvO) strategies present challenges for DP due to the need for multiple binary SVMs, leading to repeated data accesses for training samples and, consequently, a repeated consumption of the privacy budget for each classifier.

---

*Corresponding author

†Code implementation: https://github.com/JinseongP/private_multiclass_svm

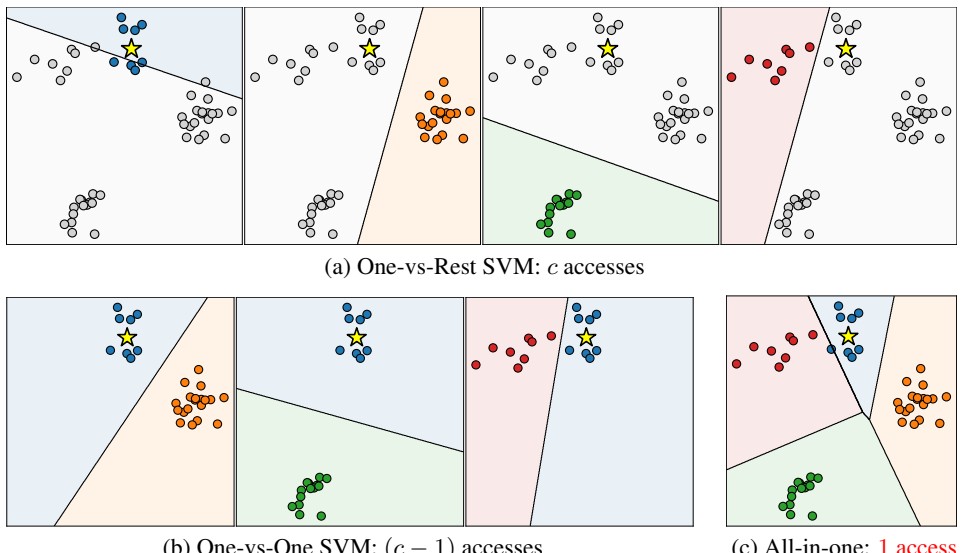

(a) One-vs-Rest SVM: $c$ accesses

(b) One-vs-One SVM: $(c-1)$ accesses

(c) All-in-one: 1 access

Figure 1: Illustration of multi-class classification strategies for $c$ classes. The individual sample ($\star$) is queried repeatedly in (a) and (b), but only once in (c). Each color represents a class.

To address the problem of multiple data accesses in support vector classification, we introduce a DP-friendly and straightforward solution to minimize the privacy cost of multi-class SVMs by leveraging all-in-one SVMs [14–17]. All-in-one SVMs solve the joint convex optimization problem, which allows for a single access to each data point while maximizing margins for multi-class classifiers. Fig. 1 demonstrates the advantages of the all-in-one method in this paper in terms of data access, compared to other strategies in multi-class scenarios.

In this paper, we propose a novel differentially Private Multi-class SVM (PMSVM), which significantly reduces privacy expenditures by accessing each data point only once, based on the all-in-one multi-class SVM framework. Our approach includes two methods to obtain a private model: (i) Weight Perturbation (WP), which adds Gaussian noise to the primal weight vector of all-in-one SVMs, and (ii) Gradient Perturbation (GP), which applies a smoothed hinge-loss approximation and introduces noise during gradient descent. In summary, the proposed PMSVM framework (i) reduces the number of data accesses per sample, (ii) thus achieves better utility, while (iii) preserving the key properties of SVMs. Empirical evaluations on benchmark multi-class datasets show that our method outperforms existing DP-SVM approaches in both accuracy and privacy-utility trade-offs, making it a practical solution for privacy-preserving multi-class machine learning models.

## 2 Backgrounds

### 2.1 Differential Privacy

Differential privacy (DP) [1, 2] establishes a mathematical framework to ensure the privacy of training data caused by changes in an individual sample, such as deletion and modification. To prevent the leakage of individual information through query responses, its formal definition is as follows:

**Definition 1.** *(Differential privacy) A randomized mechanism $\mathcal{M}$ satisfies $(\epsilon, \delta)$-differential privacy ($(\epsilon, \delta)$-DP) if, for two neighboring datasets $D, D' \in \mathcal{X}$, which differ in exactly one data sample, and for any set of possible outputs $\mathcal{O} \subseteq Range(\mathcal{M})$,*

$$Pr[\mathcal{M}(D) \in \mathcal{O}] \leq e^{\epsilon} Pr[\mathcal{M}(D') \in \mathcal{O}] + \delta. \tag{1}$$

The privacy loss is quantified by the parameter $\epsilon$, where smaller values of $\epsilon$ indicate a stronger privacy guarantee. The parameter $\delta$ represents the probability of failure for the mechanism $\mathcal{M}$.

**Definition 2** ($L_2$ Sensitivity). *For a function $f : \mathcal{D} \to \mathbb{R}^k$, the sensitivity $\Delta_f$ is defined as*

$$\Delta_f = \max_{D,D'} \|f(D) - f(D')\|_2, \tag{2}$$

*where $D$ and $D'$ differ by at most one element.*

We introduce widely used properties as remarks for DP, i.e., composition to boost the sequential application, and post-processing to preserve the privacy guarantee of outputs that are already private.

**Remark 1.** *(Composition [2]) Let $\mathcal{M}_1 : \mathcal{X} \to \mathcal{R}_1$ be an $(\epsilon_1, \delta_1)$-DP algorithm, and let $\mathcal{M}_2 : \mathcal{X} \to \mathcal{R}_2$ be an $(\epsilon_2, \delta_2)$-DP algorithm. Then, their combination $\mathcal{M}_{1,2} : \mathcal{X} \to \mathcal{R}_1 \times \mathcal{R}_2$ by the mapping: $\mathcal{M}_{1,2}(\cdot) = (\mathcal{M}_1(\cdot), \mathcal{M}_2(\cdot))$ is $(\epsilon_1 + \epsilon_2, \delta_1 + \delta_2)$-DP. For $k \geq 2$, the composition of $k$ algorithms, where each algorithm meets $(\epsilon, \delta)$, satisfies*

$$Pr[\mathcal{M}(D) \in \mathcal{O}] \leq e^{k\epsilon} Pr[\mathcal{M}(D') \in \mathcal{O}] + k\delta. \tag{3}$$

**Remark 2.** *(Post-processing [2]) If a mechanism $\mathcal{M} : \mathcal{X} \to \mathcal{R}_1$ is $(\epsilon, \delta)$-DP, for any randomized mapping $h : \mathcal{R}_1 \to \mathcal{R}_2$, $h \circ \mathcal{M} : \mathcal{X} \to \mathcal{R}_2$ is at least $(\epsilon, \delta)$-DP.*

Balle and Wang [18] proposed an analytic Gaussian mechanism to reduce the noise level of DP:

**Remark 3.** *(Analytic Gaussian Mechanism [18]) Let $f$ be a function with $L_2$ sensitivity $\Delta$. A Gaussian output perturbation mechanism $\mathcal{M}(\mathbf{x}) = f(\mathbf{x}) + \mathbf{z}$ with $\mathbf{z} \sim \mathcal{N}(0, \sigma^2\mathbf{I})$ satisfies $(\epsilon, \delta)$-DP for all $\epsilon \geq 0$, the if and only if*

$$\Phi\left(\frac{\Delta}{2\sigma} - \frac{\epsilon\sigma}{\Delta}\right) - e^\epsilon \Phi\left(-\frac{\Delta}{2\sigma} - \frac{\epsilon\sigma}{\Delta}\right) \leq \delta, \tag{4}$$

*where $\Phi$ is the cumulative distribution function of the standard normal distribution.*

## 2.2 Multi-class Support Vector Machine

**Binary Support Vector Machine**  Support Vector Machines (SVMs) [5] are a broadly used machine learning method with margin maximization for building binary classification boundaries. Consider a training dataset with $n$ samples, $\mathcal{D} = \{\mathbf{x}_i, y_i\}_{i=1}^n$, where each $\mathbf{x}_i \in \mathbb{R}^d$ is a feature vector and $y_i \in \{-1, +1\}$ its corresponding label. Then, the objective of SVM is as follows:

$$\min_{\mathbf{w}, b} \frac{1}{2} \|\mathbf{w}\|^2 + \frac{C}{n} \sum_{i=1}^n \xi_i \quad \text{s.t.} \quad y_i(\mathbf{w}^\top \mathbf{x}_i + b) \geq 1 - \xi_i, \quad \xi_i \geq 0, \forall i, \tag{5}$$

where $\mathbf{w} \in \mathbb{R}^d$ is the weight vector, $b$ is the bias term, $\xi_i$ represents the slack variables accounting for misclassifications, and $C$ is the regularization parameter controlling the trade-off between margin and errors. The Wolfe dual of Equation 5 is formulated as follows:

$$\max_{\boldsymbol{\alpha}} \quad \sum_{i=1}^n \alpha_i - \frac{1}{2} \sum_{i=1}^n \sum_{j=1}^n \alpha_i \alpha_j y_i y_j \mathbf{x}_i^\top \mathbf{x}_j \quad \text{s.t.} \quad 0 \leq \alpha_i \leq \frac{C}{n}, \quad \sum_{i=1}^n \alpha_i y_i = 0, \quad \forall i, \tag{6}$$

where $\boldsymbol{\alpha} = \{\alpha_1, \ldots, \alpha_n\} \in \mathbb{R}^n$ are the dual variables. Utilizing the Karush-Kuhn-Tucker (KKT), we can obtain the optimal parameter with $\tilde{\mathbf{w}} = \sum_{i=1}^n \tilde{\alpha}_i y_i \mathbf{x}_i$, where $\tilde{\alpha}$ are the optimal dual parameters of the convex optimization (6). Then, the binary support function is formulated as $f(\mathbf{x}) = \tilde{\mathbf{w}}^\top \mathbf{x} + b$.

**Multi-Class Support Vector Machine**  The most common way to expand binary SVM to multi-class is to use one-versus-one (OvR) or one-versus-one (OvO) approaches by training each classifier for each class pair or each class versus the rest, respectively. After training multiple binary classification models, we can make a decision $\tilde{y} = \arg\max_{k \in [c]} \{\mathbf{w}_k^\top \mathbf{x} + b_k\}$ for $c$ classes and class-wise weights $\mathbf{w}_k$ and bias $b_k$. Stacking the class–wise weights gives a weight matrix $W = [\mathbf{w}_1, \ldots, \mathbf{w}_c] \in \mathbb{R}^{d \times c}$ and biases $\mathbf{b} \in \mathbb{R}^c$ for multi-class classification.

Instead of calculating support vectors for each binary classification, a line of work investigated training multi-class SVM [15–17] at once, which is called all-in-one SVM methods [14]. Each all-in-one method has its own design for defining the margin. Among them, Weston and Watkins [15] (WW-SVM) formulated a multi–class classification as a single joint optimization problem:

$$\min_{W, \mathbf{b}} \quad \frac{1}{2} \sum_{k=1}^c \|\mathbf{w}_k\|_2^2 + \frac{C}{n} \sum_{i=1}^n \sum_{k \neq y_i} \xi_{ki} \tag{7}$$

$$\text{s.t.} \quad \mathbf{w}_{y_i}^\top \mathbf{x}_i + b_{y_i} \geq \mathbf{w}_k^\top \mathbf{x}_i + b_k + 1 - \xi_{ki}, \xi_{ki} \geq 0, k \in [c], k \neq y_i, i \in [n].$$

Table 1: Comparison of multi-class SVM strategies ($c$: # of classes, $n$: # of training samples).

| Method | Loss function | # variables per classifier | # classifiers | # accesses per sample |
|--------|---------------|----------------------------|---------------|------------------------|
| OvO | pair-wise QP (convex) | $2n/c$ | $c(c-1)/2$ | $c-1$ |
| OvR | class-wise QP (convex) | $n$ | $c$ | $c$ |
| All-in-one | joint QP (convex) | $nc^{\dagger}$ | $1$ | $1$ |

$\dagger$: we note that it may depend on the implementation algorithm.

Crammer and Singer (CS-SVM) proposed to penalize only the largest violating class, not all pairs, per sample [16]. Then the optimization can be written as follows:

$$\min_{W,\mathbf{b}} \quad \frac{1}{2}\sum_{k=1}^{c}\|\mathbf{w}_k\|_2^2 + \frac{C}{n}\sum_{i=1}^{n}\xi_i \tag{8}$$

$$\text{s.t.} \quad \mathbf{w}_{y_i}^{\top}\mathbf{x}_i - \mathbf{w}_k^{\top}\mathbf{x}_i + \upsilon_{y_i,k} \geq 1 - \xi_i, \xi_i \geq 0, k \in [c], i \in [n].$$

where $\upsilon_{y_i,k}$ equals to 1 if $k = y_i$ and 0 otherwise.

More recently, Nie et al. [17] developed a concept of maximizing minimum margin SVM ($M^3$-SVM), not just maximizing the margin between class pairs. With the support function $f_{kl}(\mathbf{x}) = (\mathbf{w}_k - \mathbf{w}_l)^{\top}\mathbf{x} + b_k - b_l$ between class $k$ and $l$ for $k < l$, its objective function with $L_2$-norm is

$$\min_{W,\mathbf{b}}\frac{1}{2}\sum_{k<l}\|\mathbf{w}_k - \mathbf{w}_l\|_2^2 + \frac{C}{n}\sum_{i=1}^{n}\sum_{k<l}\xi_{ikl} \tag{9}$$

$$\text{s.t.} \begin{cases} f_{kl}(\mathbf{x}_i) \geq 1 - \xi_{ikl}, & \text{for } y_i = k, \\ f_{kl}(\mathbf{x}_i) \leq -1 + \xi_{ikl}, & \text{for } y_i = l, \end{cases} \quad \xi_{ikl} \geq 0, k < l, i \in [n].$$

In summary, WW-SVM enforces c pair-wise constraints per sample, each with its own slack $\xi_{ki}$; CS-SVM imposes a single constraint by penalizing only the most-violating class and therefore uses a single slack $\xi_i$; whereas $M^3$-SVM simultaneously applies all pair-wise constraints per sample, introducing a distinct slack $\xi_{ikl}$ for every class pair.

## 3 Differentially Private Multi-Class SVM

### 3.1 Motivation: Advantages of All-in-One SVM for Privacy

We begin by comparing the trade-offs of multi-class SVM strategies in Table 1. Each method has its own strengths. For instance, the OvO approach requires only $2N/c$ dual variables per problem, which allows it to scale efficiently and be easily parallelized. The OvR method grows linearly with the number of classes, avoiding the quadratic explosion at inference time. The primary advantage of the all-in-one SVM method is its ability to build a robust classifier in one step by calculating the pair-wise or maximum margin at once, compared to an ensemble of binary classifiers that may become overfitted to each individual class [17]. Note that efficiency improvements may vary depending on the specific implementation of each algorithm.

When focusing on privacy, we observe that all-in-one SVMs have a clear advantage. They reduce the number of data accesses needed to build classifiers. In contrast, binary classification methods such as OvO or OvR require multiple accesses to data samples, which increase linearly with the number of classes $c$, consuming the privacy budget repeatedly with each access. Specifically, the composition theorem (Remark 1) states that the number of accesses to individual data samples directly affects the noise level if each mechanism has a dependency on training data. When applying this principle to the OvR case, the composition of c classifiers requires $c\epsilon$ privacy budget when each binary classifier requires $\epsilon$. Therefore, to maintain the same total privacy budget, we can only allocate $\frac{\epsilon}{c}$ to each binary classifier, amplifying the amount of noise on each classifier.

In the following subsections, we propose differentially Private Multi-class SVM (PMSVM) with weight and gradient perturbations tailored for all-in-one SVMs that can significantly lower the privacy cost of building DP classifiers. In contrast to existing OvO or OvR methods, the proposed PMSVM requires only one data access to build a multi-class classifier, allowing us to utilize the full privacy budget $\epsilon$.

## 3.2 Weight Perturbation for All-in-one PMSVM

Motivated by the DP empirical risk minimization (ERM) methods [3, 8–10], we first propose a weight perturbation method for PMSVM (PMSVM-WP). In DP ERM problems, we estimate the optimal weight $\tilde{\mathbf{w}}$ and protect it by adding random noise proportional to the sensitivity. The sensitivity in Definition 2 indicates how significantly the weight can vary with the worst-case changes to individual data points. The Wolf dual problem of the all-in-one SVMs can be unified as follows [14]:

$$\min_{\boldsymbol{\alpha}} \frac{1}{2} \sum_{i,p} \sum_{j,q} M_{y_i,p,y_j,q} \, \mathbf{x}_i^\top \mathbf{x}_j \, \alpha_{i,p} \, \alpha_{j,q} - \sum_{i,p} \alpha_{i,p}$$

$$\text{s.t. } 0 \le \alpha_{i,p} \le \frac{C}{n}, \qquad \sum_{p \in P_{y_i}} \alpha_{i,p} \le \frac{C}{n}, \forall i \in [n], \quad \sum_{i=1}^{n} \sum_{p \in P_{y_i}} \alpha_{i,p} \, \nu_{y_i,p,k} = 0, \quad \forall k \in [c], \tag{10}$$

where $P_{y_i} = Y \setminus \{y_i\}$ is the set of non-true class indices for sample $i$; $\nu_{y_i,p,k} = e_{y_i,k} - e_{p,k}$ with $e_{s,k}$ denoting the $k$-th component of the basis vector $\mathbf{e}_s \in \mathbb{R}^c$; and $M_{y_i,p,y_j,q} = \sum_{k=1}^{c} \nu_{y_i,p,k} \, \nu_{y_j,q,k}$.

The convexity of convex quadratic optimization problems remains in the dual formulation of all-in-one SVMs. Therefore, we take a closer look at the leave-one-out method [19] of support vector classifiers, which bounds the difference of the support function after changing one individual sample. To calculate the sensitivity of optimal weights and support functions, we need to track the sensitivity in the dual function since the dual variables of SVM are defined per data sample.

**Definition 3** (Weight Perturbation). $\hat{\mathbf{w}} = \tilde{\mathbf{w}} + \mathbf{z}$, where $\mathbf{z} \sim \mathcal{N}(0, \sigma_{\mathbf{w}}^2 \mathbf{I})$ for optimal weight $\tilde{\mathbf{w}}$.

To calculate the sensitivity of $\tilde{\mathbf{w}}$ for DP, we derive a new Lemma, a multi-class extension of the leave-one-out bound of SVM [19], as follows:

**Lemma 1.** For a convex function $T$, a dataset $D$, and input scaler $g(\cdot)$, let $\tilde{\mathbf{w}}_D = \sum_{i=1}^{n} \tilde{\alpha}_i g(\mathbf{x}_i)$, where $(\tilde{\alpha}_1, \ldots, \tilde{\alpha}_n)$ is the solution to:

$$\min_{\boldsymbol{\alpha}} \left( \frac{1}{2} \sum_{i,p} \sum_{j,q} \sum_{k} \alpha_{i,p} \alpha_{j,q} \nu_{y_i,p} \nu_{y_j,q} g(\mathbf{x}_i)^T g(\mathbf{x}_j) + \sum_{i,p} T(-\alpha_{i,p}) \right)$$

Let $D^n$ be $D$ with the $n$-th point $\mathbf{x}_n$ removed, and let $\tilde{\mathbf{w}}_{D^n}$ be defined similarly. Then the difference of the weights between original and leave-one-out SVMs is bounded as:

$$\sum_{k=1}^{c} \|\mathbf{w}_k^{[n]} - \mathbf{w}_k\|^2 \le \lambda_{\max}(G) \|\tilde{\alpha}_n\|^2 \|g(\mathbf{x}_n)\|^2.$$

Using this Lemma, we can calculate the sensitivity of the weights $\tilde{\mathbf{w}}$ of all-in-one SVMs.

**Theorem 1** (DP guarantee of weight perturbation). $\hat{\mathbf{w}} = \tilde{\mathbf{w}} + \mathbf{z}$ (Definition 3) satisfies an $(\epsilon, \delta)$-DP when $\mathbf{z} \sim \mathcal{N}(0, \sigma_{\mathbf{w}}^2 \mathbf{I})$. For $\sigma_{\mathbf{w}}$ in Remark 4, the sensitivity of the all-in-one SVM weight $\Delta_{\mathbf{w}}$ is:

$$\Delta_{\mathbf{w}} = \frac{2C}{n} \sqrt{\lambda_{\max}(G)}, \qquad G_{pq} = \langle \nu_{y,p}, \nu_{y,q} \rangle, \tag{11}$$

where $\lambda_{\max}$ is the largest eigenvalue of the Gram matrix $G$. The support function $\hat{f}(\mathbf{x}) = \arg\max_{k \in [c]} \{\hat{\mathbf{w}}_k^\top \mathbf{x}\}$ is also $(\epsilon, \delta)$-DP.

Detailed proofs of Lemma 1 and Theorem 1 are provided in Appendix B. However, this is a generalized version of the weight perturbation in a binary setting [11, 13]. We can obtain the same sensitivity of binary support vectors in [13] with Equation 11, where $\nu_{y,p} = e_p$, thus $\Delta_{\tilde{\mathbf{w}}} = 2C/n$ in $L_2$ norm with normalization to $\max(\|g(\cdot)\|_2) = 1$. This is a tightened version of $\Delta_{\tilde{\mathbf{w}}} = 4C/n$ in binary SVM [11]. Within all-in-one SVMs, we primarily focus on CS-SVM due to the ease of calculating $\lambda_{\max}(G)$, i.e., $\sqrt{\lambda_{\max}(G)} = \sqrt{2}$ due to $\nu_{y,p} = e_y - e_p$. Therefore, we can expand the sensitivity of binary weight to multi-class weight at a cost of $\sqrt{2}$ ratio while reducing the access to training data regardless of the class numbers, which gives a significant advantage for the multi-class scenario, when $c > 2$.

## 3.3 Gradient Perturbation for All-in-One PMSVM

In addition to solving the SVM dual solution through weight perturbation, we now focus on the primal solution and utilize a smoothed approximation of the hinge loss to compute gradients. Since gradient methods outperform output perturbation methods [8], we refer to our approach as gradient perturbation for PMSVM (PMSVM-GP). Specifically, Nie et al. [17] proposed a smoothed version of Equation 9 for gradient updates in all-in-one SVMs, introducing a small perturbation $\varsigma \geq 0$:

$$\min_{W,\mathbf{b}} \sum_{i=1}^{n} \sum_{k \neq y_i} \frac{\gamma_{ik} + \sqrt{\gamma_{ik}^2 + \varsigma^2}}{2} + \frac{C}{n} \sum_{k<l} \|\mathbf{w}_k - \mathbf{w}_l\|_2^2 + \mu(\|W\|_F^2 + \|\mathbf{b}\|_2^2), \qquad (12)$$

where $\gamma_{ik} = 1 - \left(\mathbf{w}_{y_i}^\top x_i + b_{y_i} - \mathbf{w}_k^\top \mathbf{x}_i - b_k\right)$ is replaced by the smooth approximation $g_\varsigma(\gamma_{ik}) = \left(\gamma_{ik} + \sqrt{\gamma_{ik}^2 + \varsigma^2}\right)/2$ for $\varsigma > 0$. $\mu$ is a small regularization parameter to ensure a unique solution.

**Definition 4** (Gradient Perturbation). $\hat{\mathbf{w}}_{t+1} = \hat{\mathbf{w}}_t - \eta_t \hat{\mathbf{g}}_t = \hat{\mathbf{w}}_t - \eta_t[\mathcal{M}_t(\mathbf{w}_t, \mathcal{D}) + \mathbf{z}]$ *where* $\mathbf{z}_t \sim \mathcal{N}(0, \sigma_{\mathbf{w}_t}^2 \mathbf{I})$ *for update step* $t \in [0, \ldots, T-1]$ *with gradient update mechanism* $\mathcal{M}_t$ *and learning rate* $\eta_t$. *The final weight of the gradient update is* $\hat{\mathbf{w}} = \hat{\mathbf{w}}_T$.

Due to the strong convexity of (12), following the proof of [17] and the positive definiteness of its Hessian matrix, the convergence of the loss function with gradient methods is guaranteed.

**Lemma 2.** *(Moments accountant [20]). There exist constant* $c_1$ *and* $c_2$ *so that given total steps* $T$ *and sampling probability* $q$, *for any* $\epsilon < c_1 q^2 T$, *gradient updates guarantee* $(\epsilon, \delta)$-*DP, for any* $\delta > 0$ *if we choose*

$$\sigma \geq c_2 \frac{q\sqrt{T \log(1/\delta)}}{\epsilon}. \qquad (13)$$

To ensure the gradient updates are private, we use differentially private gradient descent (DP-GD) or its mini-batch stochastic version, differentially private stochastic gradient descent (DP-SGD). For updates, we should choose the noise level $\sigma$ with the privacy budget $(\epsilon, \delta)$ as follows:

**Theorem 2** (DP guarantee of gradient perturbation). $\hat{\mathbf{w}}_{t+1} = \hat{\mathbf{w}}_t - \eta_t[\mathcal{M}_t(\mathbf{w}_t, \mathcal{D}) + \mathbf{z}_t]$ *(Definition 4) and its final weight* $\tilde{\mathbf{w}}_T$ *satisfy* $(\epsilon, \delta)$-*DP when updating as follows:*

$$\hat{\mathbf{w}}_{t+1} = \hat{\mathbf{w}}_t - \eta_t \hat{\mathbf{g}}_t = \hat{\mathbf{w}}_t - \eta_t \left\{ \frac{1}{n} \sum_{i=1}^{n} \frac{\nabla^{(t)}(\mathbf{x}_i)}{\max(1, \|\nabla^{(t)}(\mathbf{x}_i)\|_2/R)} + \mathbf{z}_t \right\}, \qquad (14)$$

*where individual gradients of* $\mathbf{x}_i$, $\nabla^{(t)}(\mathbf{x}_i) := \left[\nabla_1^{(t)}(\mathbf{x}_i), \ldots, \nabla_c^{(t)}(\mathbf{x}_i)\right]$, *are calculated as:*

$$\nabla_k^{(t)} = \begin{cases} -\sum_{l \neq k} \dfrac{\gamma_{il} + \sqrt{\gamma_{il}^2 + \varsigma^2}}{2\sqrt{\gamma_{il}^2 + \varsigma^2}} \mathbf{x}_i + 2\lambda \sum_{l \neq k}(\mathbf{w}_k - \mathbf{w}_l) + 2\mu \mathbf{w}_k, & k = y_i, \\[4mm] \dfrac{\gamma_{ik} + \sqrt{\gamma_{ik}^2 + \varsigma^2}}{2\sqrt{\gamma_{ik}^2 + \varsigma^2}} \mathbf{x}_i + 2\lambda \sum_{l \neq k}(\mathbf{w}_k - \mathbf{w}_l) + 2\mu \mathbf{w}_k, & k \neq y_i, \end{cases} \qquad (15)$$

*and* $\mathbf{z}_t \sim \mathcal{N}(0, R^2 \sigma^2 \mathbf{I})$ *with the* $\sigma$ *in Lemma 2 and individual gradient clipped to size* $R$.

Refer to the appendix of [20] for the proof of the moments accountant of DP gradient methods, while we calculate the gradients following Equation 15. To show the advantage of the proposed method by reducing the noise level, we now investigate the utility gain of our method compared to previous DP-SVMs in the same privacy budget. As the objective function is strictly convex, we can guarantee a tight error bound [9, 21] with gradient updates as follows:

**Lemma 3.** *([21]) Suppose* $F(w)$ *is* $\lambda$-*strongly convex and let* $\tilde{\mathbf{w}} = \arg\min_{\mathbf{w}} F(\mathbf{w})$. *Consider the stochastic gradient update*

$$\mathbf{w}_{t+1} = \mathbf{w}_t - \eta_t[\mathcal{M}_t(\mathbf{w}_t, \mathcal{D})]$$

*where* $\mathbb{E}[\mathcal{M}_t(\mathbf{w}_t), \mathcal{D}] = \nabla F(\mathbf{w}_t)$, $\mathbb{E}[\|\mathcal{M}_t(\mathbf{w}_t)\|_2^2] \leq G^2$, *and the learning rate schedule is* $\eta_t = \frac{1}{\lambda t}$. *Then, for any* $T > 1$,

$$\mathbb{E}[F(\mathbf{w}_T) - F(\tilde{\mathbf{w}})] = \mathcal{O}\left(\frac{G^2 \log(T)}{\lambda T}\right).$$

Then, by $\tau \in (0, 1]$ is the ratio of $\sigma$ in the Gaussian noise of ours to that of the OvR and OvO settings, i.e., $\tau = \sigma_{\text{ours}}/\sigma_{\text{OvR}}$ or $\tau = \sigma_{\text{ours}}/\sigma_{\text{OvO}}$, we can make a tighter bound of convergences of the noised gradient update $\mathbf{w}_T^{(\tau)}$ compared to non-private gradient update $\mathbf{w}_T$ as follows:

**Theorem 3** (Utility Advantage). *Let each single–example loss $f(w, z_i)$ be L–Lipschitz and the population objective $F(w) = \mathbb{E}_z[f(w, z)]$ be $\lambda$–strongly convex. Consider the noisy gradient update*

$$\mathbf{w}_{t+1} = \hat{\mathbf{w}}_t - \eta_t \hat{\mathbf{g}}_t = \mathbf{w}_t - \eta_t \Big( n \nabla f(\mathbf{w}_t, \mathcal{D}) + \mathbf{z}_\tau \Big), \qquad \mathbf{z}_\tau \sim \mathcal{N}\big(0, \tau^2 \sigma^2 \mathbf{I}\big),$$

*where $\tau \in (0, 1]$ is the ratio of $\sigma$ in the Gaussian noise. Let $\mathbf{w}_T^{(\tau)}$ be the T-th iteration with noise $\mathbf{z}_\tau$ and $\mathbf{w}_T$ is without noise addition. Then, with a decayed learning rate $\eta(t) = \frac{1}{\lambda t}$, for any $T > 1$,*

$$\mathbb{E}\big[F(\mathbf{w}_T) - F\big(\mathbf{w}_T^{(\tau)}\big)\big] = \mathcal{O}\Big( \frac{d\sigma^2 \big(1 - \tau^2\big) \log T}{\lambda T} \Big).$$

*Instead, with a constant learning rate $\eta(t) = \eta = \frac{c}{\lambda}$ with $0 < c \le \frac{1}{2}$, for any $T > 1$,*

$$\mathbb{E}\big[F(\mathbf{w}_T) - F\big(\mathbf{w}_T^{(\tau)}\big)\big] = \mathcal{O}\Big( c \, d\sigma^2 \big(1 - \tau^2\big) \Big).$$

Thus, we theoretically prove that the reduced noise level in the all-in-one classifier results in smaller error compared to non-private updated points.

The main reason to use clipping-based gradient updates in DP-SGD is that these approaches are extensively studied, allowing us to leverage recent analytical techniques for gradient-based optimization. By the moments accountant, we can lower the privacy cost of the composition to $(O(q\epsilon\sqrt{T}), \delta)$-DP [20]. Also, we can utilize the Poisson sub-sampling [22] for mini-batch update, since private optimization has limited access to the data samples due to the privacy-utility trade-offs. Furthermore, adopting advanced techniques is feasible within our framework. For example, for stable convergence, we can adapt adaptive moment methods, such as Adam [23] and its DP variants [24], update the weight of Equation 14 into adaptive gradient perturbation (AGP) as follows:

$$\hat{\mathbf{m}}_t = \beta_1 \hat{\mathbf{m}}_{t-1} + (1 - \beta_1)\,\hat{\mathbf{g}}_t, \; \hat{\mathbf{v}}_t = \beta_2 \hat{\mathbf{v}}_{t-1} + (1 - \beta_2)(\hat{\mathbf{g}}_t \odot \hat{\mathbf{g}}_t),$$
$$\hat{\mathbf{m}}_t = \frac{\hat{\mathbf{m}}_t}{1 - \beta_1^t}, \; \hat{\mathbf{v}}_t = \frac{\hat{\mathbf{v}}_t}{1 - \beta_2^t}, \; \hat{\mathbf{w}}_t = \hat{\mathbf{w}}_{t-1} - \eta \frac{\hat{\mathbf{m}}_t}{\sqrt{\hat{\mathbf{v}}_t} + \gamma}, \tag{16}$$

with the gradient momentum $\hat{\mathbf{m}}_t$ and the second-moment accumulator $\hat{\mathbf{v}}_t$ of Equation 15 as in [24], which helps to reduce the iteration complexity. Because of the post-processing in Remark 2, Equation 16 still guarantees $(\epsilon, \delta)$-DP guarantee of Lemma 2.

## 4 Related Works

Existing DP-SVM methods primarily focus on binary classification tasks. Chaudhuri et al. [3] investigated the use of DP convex optimization in SVMs, and Rubinstein et al. [11] expanded the methods with kernels with weight perturbation in binary setups. As the support vectors of dual formulation are coupled with the subset of training data, Jain and Thakurta [12] published the private weight based on the interactive scenario of model users. Ding et al. [9] used the gradient method for SVM with smoothed loss. None of the following works on DP-SVMs [25, 26] investigated the privacy amplification of multi-class SVMs.

Park et al. [13] argued a similar research question to our paper that multi-class SVMs need multiple accesses for training. Rather than mitigating within the boundary of SVMs, they detoured from the method with a kernel clustering and labeling method. On the other hand, we directly utilized the all-in-one SVMs to reduce the number of data accesses, which is compatible with the non-DP methods, such as CS-SVM or $M^3$-SVM.

## 5 Experiments

### 5.1 Experimental Design

**Datasets** We used multi-class classification datasets from the University of California at Irvine (UCI) repository [27] for various data types: Cornell (CS web pages), Dermatology (clinical skin

Table 2: Performance comparison across datasets for weight- and gradient-perturbation methods. We **bold** the best accuracy within each perturbation strategy.

| Data | $\epsilon$ | Weight Perturbation | | | Gradient Perturbation | | | |
|---|---|---|---|---|---|---|---|---|
| | | PrivateSVM [11] | OPERA [9] | PMSVM-WP | GRPUA [9] | Linear [20] | PMSVM-GP | PMSVM-AGP |
| Cornell | 1 | $0.197 \pm 0.089$ | $0.244 \pm 0.095$ | $\mathbf{0.599} \pm 0.199$ | $0.493 \pm 0.029$ | $0.624 \pm 0.035$ | $0.623 \pm 0.018$ | $\mathbf{0.693} \pm 0.032$ |
| | 2 | $0.278 \pm 0.086$ | $0.333 \pm 0.127$ | $\mathbf{0.730} \pm 0.242$ | $0.572 \pm 0.011$ | $0.695 \pm 0.033$ | $0.695 \pm 0.032$ | $\mathbf{0.707} \pm 0.023$ |
| | 4 | $0.448 \pm 0.139$ | $0.505 \pm 0.172$ | $\mathbf{0.761} \pm 0.248$ | $0.692 \pm 0.043$ | $0.747 \pm 0.010$ | $0.723 \pm 0.026$ | $\mathbf{0.752} \pm 0.023$ |
| | 8 | $0.597 \pm 0.201$ | $0.683 \pm 0.222$ | $\mathbf{0.770} \pm 0.250$ | $0.746 \pm 0.023$ | $\mathbf{0.792} \pm 0.015$ | $0.789 \pm 0.024$ | $0.765 \pm 0.024$ |
| Dermatology | 1 | $0.240 \pm 0.120$ | $0.296 \pm 0.131$ | $\mathbf{0.711} \pm 0.098$ | $0.787 \pm 0.041$ | $\mathbf{0.911} \pm 0.028$ | $0.865 \pm 0.050$ | $0.905 \pm 0.017$ |
| | 2 | $0.422 \pm 0.142$ | $0.465 \pm 0.141$ | $\mathbf{0.821} \pm 0.076$ | $0.903 \pm 0.026$ | $0.930 \pm 0.018$ | $\mathbf{0.954} \pm 0.021$ | $0.951 \pm 0.042$ |
| | 4 | $0.595 \pm 0.146$ | $0.698 \pm 0.134$ | $\mathbf{0.894} \pm 0.064$ | $0.968 \pm 0.015$ | $0.970 \pm 0.022$ | $0.965 \pm 0.015$ | $\mathbf{0.978} \pm 0.012$ |
| | 8 | $0.858 \pm 0.078$ | $0.897 \pm 0.058$ | $\mathbf{0.923} \pm 0.052$ | $0.976 \pm 0.015$ | $0.973 \pm 0.014$ | $0.970 \pm 0.018$ | $\mathbf{0.976} \pm 0.018$ |
| HHAR | 1 | $0.575 \pm 0.137$ | $0.674 \pm 0.105$ | $\mathbf{0.889} \pm 0.013$ | $0.851 \pm 0.020$ | $0.887 \pm 0.005$ | $0.908 \pm 0.008$ | $\mathbf{0.929} \pm 0.007$ |
| | 2 | $0.789 \pm 0.101$ | $\mathbf{0.864} \pm 0.040$ | $0.896 \pm 0.007$ | $0.861 \pm 0.013$ | $0.920 \pm 0.006$ | $0.944 \pm 0.002$ | $\mathbf{0.946} \pm 0.004$ |
| | 4 | $0.889 \pm 0.023$ | $\mathbf{0.898} \pm 0.016$ | $0.898 \pm 0.006$ | $0.873 \pm 0.013$ | $0.936 \pm 0.003$ | $\mathbf{0.958} \pm 0.004$ | $0.956 \pm 0.006$ |
| | 8 | $0.912 \pm 0.009$ | $\mathbf{0.913} \pm 0.005$ | $0.898 \pm 0.006$ | $0.869 \pm 0.006$ | $0.949 \pm 0.002$ | $\mathbf{0.962} \pm 0.003$ | $0.959 \pm 0.003$ |
| ISOLET | 1 | $0.053 \pm 0.021$ | $0.046 \pm 0.020$ | $\mathbf{0.262} \pm 0.103$ | $0.060 \pm 0.020$ | $0.466 \pm 0.042$ | $0.442 \pm 0.011$ | $\mathbf{0.501} \pm 0.025$ |
| | 2 | $0.054 \pm 0.017$ | $0.063 \pm 0.023$ | $\mathbf{0.502} \pm 0.075$ | $0.078 \pm 0.023$ | $0.672 \pm 0.038$ | $0.670 \pm 0.022$ | $\mathbf{0.687} \pm 0.017$ |
| | 4 | $0.072 \pm 0.032$ | $0.123 \pm 0.044$ | $\mathbf{0.699} \pm 0.056$ | $0.117 \pm 0.012$ | $\mathbf{0.820} \pm 0.014$ | $0.812 \pm 0.023$ | $0.804 \pm 0.010$ |
| | 8 | $0.137 \pm 0.048$ | $0.205 \pm 0.055$ | $\mathbf{0.813} \pm 0.031$ | $0.197 \pm 0.038$ | $0.858 \pm 0.024$ | $\mathbf{0.874} \pm 0.009$ | $0.840 \pm 0.013$ |
| USPS | 1 | $0.184 \pm 0.071$ | $0.236 \pm 0.068$ | $\mathbf{0.884} \pm 0.018$ | $0.747 \pm 0.018$ | $0.875 \pm 0.009$ | $0.879 \pm 0.005$ | $\mathbf{0.897} \pm 0.006$ |
| | 2 | $0.257 \pm 0.093$ | $0.367 \pm 0.105$ | $\mathbf{0.919} \pm 0.008$ | $0.845 \pm 0.007$ | $0.904 \pm 0.009$ | $\mathbf{0.911} \pm 0.006$ | $0.907 \pm 0.006$ |
| | 4 | $0.503 \pm 0.121$ | $0.642 \pm 0.088$ | $\mathbf{0.925} \pm 0.007$ | $0.876 \pm 0.005$ | $\mathbf{0.922} \pm 0.005$ | $0.920 \pm 0.003$ | $0.917 \pm 0.002$ |
| | 8 | $0.769 \pm 0.069$ | $0.843 \pm 0.026$ | $\mathbf{0.929} \pm 0.006$ | $0.880 \pm 0.004$ | $0.928 \pm 0.005$ | $\mathbf{0.930} \pm 0.001$ | $0.924 \pm 0.003$ |
| Vehicle | 1 | $0.312 \pm 0.058$ | $\mathbf{0.331} \pm 0.053$ | $0.281 \pm 0.070$ | $0.568 \pm 0.052$ | $0.661 \pm 0.046$ | $0.620 \pm 0.050$ | $\mathbf{0.696} \pm 0.060$ |
| | 2 | $\mathbf{0.356} \pm 0.073$ | $0.345 \pm 0.053$ | $0.307 \pm 0.064$ | $0.659 \pm 0.055$ | $0.722 \pm 0.034$ | $0.676 \pm 0.056$ | $\mathbf{0.753} \pm 0.007$ |
| | 4 | $0.377 \pm 0.064$ | $\mathbf{0.384} \pm 0.068$ | $0.378 \pm 0.097$ | $0.728 \pm 0.012$ | $0.711 \pm 0.035$ | $0.707 \pm 0.018$ | $\mathbf{0.733} \pm 0.023$ |
| | 8 | $\mathbf{0.386} \pm 0.057$ | $0.379 \pm 0.047$ | $0.478 \pm 0.106$ | $0.722 \pm 0.020$ | $0.729 \pm 0.024$ | $0.721 \pm 0.063$ | $\mathbf{0.766} \pm 0.009$ |

records), HHAR (wearable activity sensors), ISOLET (spoken alphabet), USPS (hand-written digits), and Vehicle (vehicle silhouettes).

**Baselines** For comparison methods, we compared with both existing weight and gradient perturbation methods in DP-SVMs based on OVR strategies. For weight perturbation, we compared with PrivateSVM [11] and OPERA [9]. For gradient methods, we compare with GRPUA [9]. Additionally, for gradient descent [20] for a neural network classification, we used a linear layer (Linear), with the cross-entropy loss, which shares the same architecture but with the loss used in neural network classification. We exclude the DP-SVM models having interaction with users [12] and local DP [25].

**Experimental details** For privacy budget, we fixed $\delta = 10^{-5}$ on various $\epsilon$. We reported the mean and standard deviation on each setting, where we used 20 runs for weight perturbation and 5 runs for gradient perturbations. We performed a grid search on each method to find the well-performing one on $\epsilon = 4$, and used the obtained parameters for each model on other epsilons. We searched on $C/n$ for weight perturbation, and learning rate $\eta_t$, gradient steps $T$, and fixed the clipping $R = 1$ for gradient methods. We further utilize the min-max scaler for weight perturbation to bound the input sensitivity to 1 and thus calculate the sensitivity of $\tilde{\mathbf{w}}$ easily. We utilize a Poisson sub-sampling batch size of 128 for gradient methods.

We utilized the SVM packages in Sklearn [28] for weight perturbation, and the Opacus [29] for gradient descent methods based on Pytorch [30]. All experiments were run on an Intel(R) Xeon(R) CPU E5-2680 v3 @ 2.50GHz and a single NVIDIA GeForce RTX 4090.

Code is available at `https://github.com/JinseongP/private_multiclass_svm`. Refer to Appendix C for further details of datasets and experimental settings.

## 5.2 Classification Results

Table 2 presents the multi-class classification results of weight and gradient perturbation methods. In both perturbation strategies, our method, based on all-in-one SVM, surpasses previous SVM strategies in multi-class settings. For weight perturbation, our method significantly improves the performance, especially with small $\epsilon$, where the decision is more perturbed with noise, and thus reducing noise in our method gives a big potential for utility improvement. The observed underperformance on the Vehicle dataset likely stems from the poor baseline performance of the all-in-one SVM itself, as we used a uniform hyperparameter $C$ across all methods.

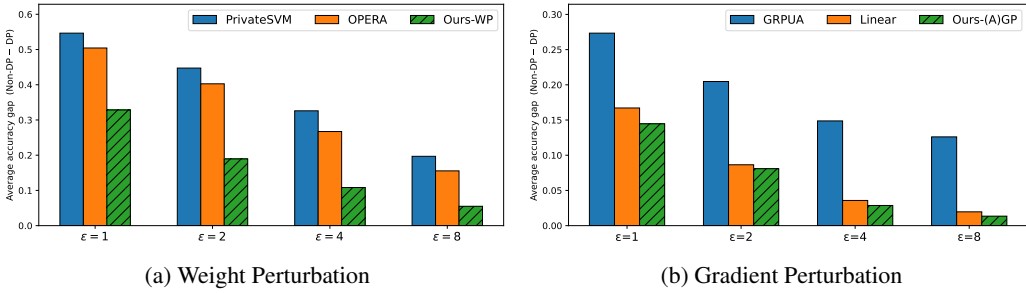

| (a) Weight Perturbation | (b) Gradient Perturbation |

Figure 2: Accuracy gap between DP-SVM methods and their non-private baselines ($\epsilon = \infty$). Lower value indicates a smaller accuracy–privacy trade-off, thus indicating a DP-friendly property.

Gradient perturbation methods typically outperform weight perturbation methods by mitigating instability, adding noise incrementally during training rather than afterward. In both the standard gradient descent and its adaptive variant, our approach exceeds the performance of the existing gradient method, GRPUA. Additionally, our margin-maximizing gradients surpass linear layers with CE loss, confirming observations by [17] within DP scenarios.

To show the DP-friendly advantages of employing an all-in-one method, we depict the accuracy gap between DP and non-DP ($\epsilon = \infty$) settings for each method in Fig. 2. Specifically, we calculate non-DP accuracy and show the average accuracy gap across datasets listed in Table 2, where the lower value has better utility-privacy trade-offs. Within a low level of privacy guarantee (higher $\epsilon$), the accuracy gap remains small (under 0.15), and differences among methods are also small. Conversely, under tighter privacy constraints (lower $\epsilon$), the accuracy gap widens significantly, emphasizing the strength of each method for DP. Consequently, the proposed PMSVM method proves to be DP-friendly and consistently robust across diverse multi-class datasets. Detailed individual dataset results are available in Appendix C.

## 5.3 Additional Experiments

We now present additional experiments concerning our proposed methods.

**Convergence** Fig. 3 shows the training loss, training accuracy, and test accuracy used to evaluate the convergence of our method. Smaller $\epsilon$ values introduce larger noise, which hinders convergence and leads to loss divergence at $\epsilon = 1$. In contrast, with larger $\epsilon$, the model effectively minimizes the loss and converges well for understanding the generalization performance. Overall, adaptive optimizers achieve faster convergence in the early training stages.

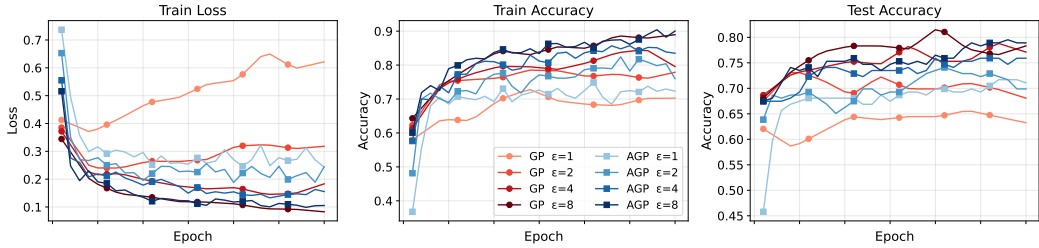

Figure 3: Convergence curves of training loss, training accuracy, and test accuracy for the proposed PMSVM-GP and PMSVM-AGP methods.

**Learning rate decay** Classical gradient-based SVMs utilize a decaying learning rate schedule [9] for convergence, while DP-based deep learning approaches often use a constant learning rate [31, 32] to reduce the number of interactions. To investigate this further, we report accuracy and absolute error under a linear learning-rate decay in Table 3. The results, including additional datasets in the Appendix, show no clear advantage for either strategy.

Table 3: Ablation study on the effect of learning rate decay for the proposed gradient perturbation methods. We bold the better performance in **bold** and Diff indicates the absolute difference w/ and w/o lr decay. Results for other datasets are shown in Appendix C.

| Dataset | $\epsilon$ | PMSVM-GP | + lr decay | Diff | PMSVM-AGP | + lr decay | Diff |
|---------|-----------|----------|------------|------|-----------|------------|------|
| Cornell | 1 | 0.663±0.010 | **0.673**±0.033 | 0.010 | **0.692**±0.025 | **0.692**±0.022 | 0.000 |
| | 2 | 0.719±0.023 | **0.743**±0.009 | 0.024 | **0.728**±0.018 | 0.706±0.021 | 0.022 |
| | 4 | **0.771**±0.014 | 0.752±0.010 | 0.019 | **0.772**±0.019 | 0.748±0.013 | 0.024 |
| | 8 | **0.770**±0.012 | 0.765±0.015 | 0.005 | 0.769±0.029 | **0.774**±0.014 | 0.005 |
| Dermatology | 1 | 0.895±0.038 | **0.908**±0.029 | 0.013 | **0.900**±0.031 | 0.824±0.065 | 0.076 |
| | 2 | **0.949**±0.022 | 0.938±0.043 | 0.011 | **0.941**±0.024 | 0.930±0.040 | 0.011 |
| | 4 | **0.973**±0.017 | 0.938±0.021 | 0.035 | **0.984**±0.006 | 0.973±0.010 | 0.011 |
| | 8 | **0.976**±0.015 | 0.957±0.026 | 0.019 | **0.984**±0.006 | **0.984**±0.006 | 0.000 |

**Computation** We then compare the computational time of existing multi-class DP-SVMs based on weight and gradient perturbation in Table 4. Because weight-perturbation baselines rely on the built-in scikit-learn implementations, their running times are essentially those of the OvR and all-in-one strategies: the Crammer–Singer formulation solves a single joint QP with $nc$ variables, whereas OvR decomposes into $c$ independent binary SVMs, each with $n$ variables. Given that standard QP solvers scale as $O(\text{num of params}^3)$, Crammer–Singer entails $O(n^3 c^3)$, while OvR requires $O(cn^3)$. In practice (e.g., in scikit-learn using LIBLINEAR), the observed gap is smaller practically. This explains the runtime gap between OPERA and PMSVM-WP, such as ISOLET. However, we highlight that the time for noise addition is negligible to ensure DP. For gradient methods, GRPUA performs $c$ separate binary classifications and therefore takes several times longer than the proposed gradient-based private SVM, which updates all parameters all at once.

Table 4: Computation time for weight and gradient perturbation methods. We measured total training time for weight perturbation methods with scikit-learn built-in SVM in seconds, and per-iteration time for gradient perturbation methods in milliseconds (ms).

| | Method | Cornell | Dermatology | HHAR | ISOLET | USPS | Vehicle | Average |
|---|--------|---------|-------------|------|--------|------|---------|---------|
| Weight | OPERA | 0.04±0.01 | 0.01±0.00 | 1.37±0.11 | 0.62±0.08 | 1.10±0.68 | 0.01±0.00 | 0.53 (sec) |
| | Ours-WP | 0.06±0.02 | 0.01±0.00 | 1.68±0.50 | 1.55±0.44 | 1.27±0.73 | 0.01±0.00 | 0.76 (sec) |
| Gradient | GRPUA | 37.57±0.73 | 24.13±0.34 | 44.90±3.03 | 86.60±5.24 | 47.47±2.25 | 19.39±1.17 | 43.34 (ms/iter) |
| | Ours-GP | 16.04±1.43 | 5.90±0.90 | 5.47±2.44 | 19.09±7.26 | 5.03±0.51 | 4.06±0.18 | 9.27 (ms/iter) |

Further datasets and detailed results, including ablation studies on clipping threshold and batch sizes, are provided in Appendix C.

# 6  Conclusion

This paper presents a novel privacy-preserving multi-class SVM framework designed to mitigate the issue of repeated data access found in existing multi-class SVM approaches under DP scenarios. By employing all-in-one methods, our framework significantly reduces the noise level through decreased data access, eliminating the need for multiple binary classifiers for both weights and gradients.
**Limitation and Social Impact:** We contribute to enhancing the trustworthiness of machine learning models through improved privacy protections. However, further experiments are necessary in domains where privacy is particularly crucial, such as healthcare, face recognition, or IoT domains.

# Acknowledgments

This research was supported by the National Research Foundation of Korea (NRF) (No. RS-2024-00338859), and the Institute of Information & communications Technology Planning & Evaluation (IITP) (No. RS-2022-II220984) grant funded by the Korean government (MSIT). Jinseong Park is supported by a KIAS Individual Grant (AP102301) via the Center for AI and Natural Sciences at Korea Institute for Advanced Study. This work was supported by the Center for Advanced Computation at Korea Institute for Advanced Study.

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

# A  Notations

We summarize the important notation used in the paper in Table 5.

Table 5: Summary of notations.

| | |
|---|---|
| **Differential Privacy** | |
| $\mathcal{M}$ | randomized mechanism |
| $D, D'$ | neighboring datasets |
| $\epsilon, \delta$ | privacy parameter |
| $\Delta_f$ | $L_2$-sensitivity of $f$ |
| $\|\cdot\|_2$ | Euclidean norm |
| **Support Vector Machine** | |
| $\mathcal{D} = \{(\mathbf{x}_i, y_i)\}_{i=1}^n$ | training set |
| $n$ | # of samples |
| $d$ | feature dimension |
| $\mathbf{x}_i \in \mathbb{R}^d$ | feature vector |
| $\mathbf{w} \in \mathbb{R}^{d \times c}$ | flattened weight vector |
| $b \in \mathbb{R}$ | bias term |
| $\xi_i \geq 0$ | slack variable |
| $C$ | regularization parameter |
| $\boldsymbol{\alpha}$ | dual variables |
| $f(\mathbf{x})$ | decision function |
| $c$ | # of classes |
| $\mathbf{w}_k \in \mathbb{R}^d, b_k \in \mathbb{R}$ | class–wise weight & bias |
| $W = [\mathbf{w}_1, \ldots, \mathbf{w}_c] \in \mathbb{R}^{d \times c}$ | weight matrix |
| $\mathbf{b} = [b_1, \ldots, b_c]^\top \in \mathbb{R}^c$ | bias vector |
| $\alpha_{i,p}$ | dual var for sample $i$, class $p$ |
| $P_{y_i} = Y \setminus \{y_i\}$ | non–true class indices |
| $\nu_{y_i,p,k} = e_{y_i,k} - e_{p,k}$ | encoding vector |
| $M_{y_i,p,y_j,q} = \sum_{k=1}^c \nu_{y_i,p,k}\, \nu_{y_j,q,k}$ | interaction matrix |
| **Methodology** | |
| $\tilde{\mathbf{w}}$ | optimal primal weight |
| $\hat{\mathbf{w}} = \tilde{\mathbf{w}} + \mathbf{z}, \mathbf{z} \sim \mathcal{N}(0, \sigma_{\mathbf{w}}^2 I)$ | noisy weight |
| $G_{pq} = \langle \nu_{y,p}, \nu_{y,q} \rangle$ | Gram matrix |
| $\lambda, \lambda_{\max}$ | convexity, top eigenvalue |
| $\eta_t$ | learning rate |
| $R$ | clipping norm bound |
| $\sigma \geq c_2 \frac{q\sqrt{T \ln(1/\delta)}}{\epsilon}$ | noise scale (moments accountant) |
| $\gamma_{ik} = 1 - \left(\mathbf{w}_{y_i}^\top x_i + b_{y_i} - \mathbf{w}_k^\top x_i - b_k\right)$ | margin violation |
| $\varsigma$ | smoothing parameter |
| $q$ | sampling prob. in minibatch |
| $T$ | total update steps |
| $\tau \in (0, 1]$ | noise scaling factor |

# B  Proofs

## B.1  Proof of Theorem 1

**(Restated) Lemma 1.** For a convex function $T$, a dataset $D$, and input scaler $g(\cdot)$, let $\tilde{\mathbf{w}}_D = \sum_{i=1}^n \tilde{\alpha}_i g(\mathbf{x}_i)$, where $(\tilde{\alpha}_1, \ldots, \tilde{\alpha}_n)$ is the solution to:

$$\min_{\boldsymbol{\alpha}} \left( \frac{1}{2} \sum_{i,p} \sum_{j,q} \sum_k \alpha_{i,p} \alpha_{j,q} \nu_{y_i,p} \nu_{y_j,q} g(\mathbf{x}_i)^T g(\mathbf{x}_j) + \sum_{i,p} T(-\alpha_{i,p}) \right)$$

Let $D^n$ be $D$ with the $n$-th point $\mathbf{x}_n$ removed, and let $\tilde{\mathbf{w}}_{D^n}$ be defined similarly. Then the difference of the weights between original and leave-one-out SVMs is bounded as:

$$\sum_{k=1}^{c} \|\mathbf{w}_k^{[n]} - \mathbf{w}_k\|^2 \leq \lambda_{\max}(G)\|\tilde{\alpha}_n\|^2\|g(\mathbf{x}_n)\|^2.$$

*Proof.* Let $\tilde{\alpha}$ be the solution of Equation 10, and $\tilde{\alpha}^{[n]}$ be the solution of Equation 10, when $n$-th training element removed (WLOG). Then, following the proof of Zhang [19], taking a subgradient of $T$ at $\tilde{\alpha}_{i,p}$ with respect to $\alpha_{i,p}$, the following first-order optimality condition holds:

$$-\nabla_1 T(-\tilde{\alpha}_{i,p}) + \sum_{j,q} M_{y_i,p,y_j,q}\, \mathbf{x}_i^\top \mathbf{x}_j\, \alpha_{j,q} = 0 \quad \forall i \leq n, p \in P_{y_i} \tag{17}$$

Multiply $(\tilde{\alpha}_{i,p}^{[n]} - \tilde{\alpha}_{i,p})$ to the equation:

$$-\nabla_1 T(-\tilde{\alpha}_{i,p})(\tilde{\alpha}_{i,p}^{[n]} - \tilde{\alpha}_{i,p}) + \sum_{j,q} M_{y_i,p,y_j,q}\, \mathbf{x}_i^\top \mathbf{x}_j\, \alpha_{j,q}(\tilde{\alpha}_{i,p}^{[n]} - \tilde{\alpha}_{i,p}) = 0 \quad \forall i \leq n-1, p \in P_{y_i} \tag{18}$$

By the definition of subgradient, we have

$$-\nabla_1 T(-\tilde{\alpha}_{i,p})(\tilde{\alpha}_{i,p}^{[n]} - \tilde{\alpha}_{i,p}) \leq T(-\tilde{\alpha}_{i,p}^{[n]}) - T(-\tilde{\alpha}_{i,p}) \tag{19}$$

Therefore, we can get

$$T(-\tilde{\alpha}_i) - \sum_{j,q} M_{y_i,p,y_j,q}\, \mathbf{x}_i^\top \mathbf{x}_j\, \alpha_{j,q}(\tilde{\alpha}_{i,p}^{[n]} - \tilde{\alpha}_{i,p}) \leq T(-\tilde{\alpha}_i^{[n]}) \tag{20}$$

Then, taking summation over $i, p$:

$$\sum_{i,p}^{n-1} \left[ T(-\tilde{\alpha}_i) - \sum_{j,q} M_{y_i,p,y_j,q}\, \mathbf{x}_i^\top \mathbf{x}_j\, \tilde{\alpha}_{j,q}^n(\tilde{\alpha}_{i,p}^{[n]} - \tilde{\alpha}_{i,p}) \right] + \frac{1}{2}\sum_{i,p}^{n-1}\sum_{j,q}^{n-1} M_{y_i,p,y_j,q}\, \mathbf{x}_i^\top \mathbf{x}_j\, \tilde{\alpha}_{i,p}^{[n]}\tilde{\alpha}_{j,q}^{[n]} \tag{21}$$

$$\leq \sum_{i,p}^{n-1} T(-\tilde{\alpha}_i^{[n]}) + \frac{1}{2}\sum_{i,p}^{n-1}\sum_{j,q}^{n-1} M_{y_i,p,y_j,q}\, \mathbf{x}_i^\top \mathbf{x}_j\, \tilde{\alpha}_{i,p}^{[n]}\tilde{\alpha}_{j,q}^{[n]} \tag{22}$$

$$\leq \sum_{i,p}^{n-1} T(-\tilde{\alpha}_i) + \frac{1}{2}\sum_{i,p}^{n-1}\sum_{j,q}^{n-1} M_{y_i,p,y_j,q}\, \mathbf{x}_i^\top \mathbf{x}_j\, \tilde{\alpha}_{i,p}\tilde{\alpha}_{j,q}. \tag{23}$$

The second inequality follows from the definition of $\tilde{\alpha}^{[n]}$, as in the proof of Lemma 1. Note that since the domain of $p$ depends on $i$, we simply notate $\sum_{i=1}^{n-1}\sum_{p \in P_{y_i}}$ as $\sum_{i,p}^{n-1}$ and $\sum_{i=1}^{n}\sum_{p \in P_{y_i}}$ as $\sum_{i,p}^{n}$ (and the same with $j$ and $q$). Next, denote $\tilde{\alpha}_{n,p}^{[n]} = 0$, then,

$$\frac{1}{2}\sum_{i,p}\sum_{j,q} M_{y_i,p,y_j,q}\, \mathbf{x}_i^\top \mathbf{x}_j\, (\tilde{\alpha}_{i,p}^{[n]} - \tilde{\alpha}_{i,p})(\tilde{\alpha}_{j,q}^{[n]} - \tilde{\alpha}_{j,q}) \leq \frac{1}{2}\sum_{p}\sum_{q} M_{y_n,p,y_n,q}\, \mathbf{x}_n^\top \mathbf{x}_n \tilde{\alpha}_{n,p}\tilde{\alpha}_{n,q} \tag{24}$$

$$= \frac{1}{2}\sum_{p}\sum_{q}\sum_{k=1}^{c} \nu_{y_n,p,k}\,\nu_{y_n,q,k} \mathbf{x}_n^\top \mathbf{x}_n \tilde{\alpha}_{n,p}\tilde{\alpha}_{n,q} \tag{25}$$

$$= \frac{1}{2}\mathbf{x}_n^\top \mathbf{x}_n \sum_{p,q} \tilde{\alpha}_{n,p}\tilde{\alpha}_{n,q}\langle \nu_{y_n,p}\,\nu_{y_n,q}\rangle \tag{26}$$

$$= \frac{1}{2}\mathbf{x}_n^\top \mathbf{x}_n \tilde{\alpha}_n^\top G \tilde{\alpha}_n \leq \frac{1}{2}\|\mathbf{x}_n\|^2 \lambda_{\max}(G)\|\tilde{\alpha}_n\|^2 \tag{27}$$

The last inequality holds because the Gram matrix is PSD. Therefore,

$$\sum_{k=1}^{c} \|\mathbf{w}_k^{[n]} - \mathbf{w}_k\|^2 \leq \lambda_{\max}(G)\|\tilde{\alpha}_n\|^2\|\mathbf{x}_n\|^2. \tag{28}$$

$\square$

Using this Lemma, we can calculate the sensitivity of the weights $\tilde{\mathbf{w}}$ of all-in-one SVMs.

**(Restated) Theorem 1.** (DP guarantee of weight perturbation) $\hat{\mathbf{w}} = \tilde{\mathbf{w}} + \mathbf{z}$ (Definition 3) satisfies an $(\epsilon, \delta)$-DP when $\mathbf{z} \sim \mathcal{N}(0, \sigma_{\mathbf{w}}^2 \mathbf{I})$. For $\sigma_W$ in Remark 4, the sensitivity of the all-in-one SVM weight $\Delta_{\mathbf{w}}$ is:

$$\Delta_{\mathbf{w}} = \frac{2C}{n} \sqrt{\lambda_{\max}(G)}, \qquad G_{pq} = \langle \nu_{y,p}, \nu_{y,q} \rangle, \tag{29}$$

where $\lambda_{\max}$ is the largest eigenvalue of the Gram matrix $G$, and $\nu_{y,q} \in \mathbb{R}^c$ is a vector that $k$th component is $\nu_{y,q,k}$. Moreover, the support function $\hat{f}(\mathbf{x}) = \arg\max_{k \in [c]} \{ \tilde{\mathbf{w}}_k^\top \mathbf{x} \}$ is also $(\epsilon, \delta)$-DP.

*Proof.* Firstly, we need to find the sensitivity of $W$. Let $T(-\alpha_{i,p}) := -\alpha_{i,p}$, which is affine and therefore convex. Then, by Lemma 1, the following inequality holds:

$$\sum_{k=1}^{c} \| \mathbf{w}_k^{[n]} - \mathbf{w}_k \|^2 \leq \lambda_{\max}(G) \| \tilde{\alpha}_n \|^2 \| \mathbf{x}_n \|^2. \tag{30}$$

For $\| \mathbf{x}_n \| \leq \kappa = \max g(\mathbf{x})$, usually set $\kappa = 1$ with normalization,

$$\| W^{[n]} - W \|_F \leq \frac{C\kappa}{n} \sqrt{\lambda_{\max}(G)}. \tag{31}$$

By triangle inequality,

$$\| W_{D'} - W_D \|_F \leq \| W_D^{[n]} - W_D \|_F + \| W_{D'}^{[n]} - W_{D'} \|_F \leq \frac{2C\kappa}{n} \sqrt{\lambda_{\max}(G)}. \tag{32}$$

Therefore, for flattened weight $\mathbf{w}$ for $W$,

$$\| \mathbf{w}_{D'} - \mathbf{w}_D \|_2 = \| W_{D'} - W_D \|_F \leq \frac{2C\kappa}{n} \sqrt{\lambda_{\max}(G)}. \tag{33}$$

Adding isotropic Gaussian noise for $\sigma_{\mathbf{w}}$ in Remark 4 therefore guarantees $(\epsilon, \delta)$-DP, and the post-processing property extends the guarantee to the decision function $\hat{f}(\cdot)$. $\qquad\square$

## B.2 Proof of Theorem 3

**(Restated) Lemma 3.** ([21]) Suppose $F(w)$ is $\lambda$-strongly convex and let $\tilde{\mathbf{w}} = \arg\min_{\mathbf{w}} F(\mathbf{w})$. Consider the stochastic gradient update

$$\mathbf{w}_{t+1} = \mathbf{w}_t - \eta_t [\mathcal{M}_t(\mathbf{w}_t, \mathcal{D})]$$

where $\mathbb{E}[\mathcal{M}_t(\mathbf{w}_t), \mathcal{D}] = \nabla F(\mathbf{w}_t)$, $\mathbb{E}[\| \mathcal{M}_t(\mathbf{w}_t) \|_2^2] \leq G^2$, and the learning rate schedule is $\eta_t = \frac{1}{\lambda t}$. Then, for any $T > 1$,

$$\mathbb{E}[F(\mathbf{w}_T) - F(\tilde{\mathbf{w}})] = \mathcal{O}\left( \frac{G^2 \log(T)}{\lambda T} \right).$$

**(Restated) Theorem 3.** *(Utility Advantage) Let each single–example loss $f(w, z_i)$ be $L$–Lipschitz and the population objective $F(w) = \mathbb{E}_z[f(w, z)]$ be $\lambda$–strongly convex. Consider the noisy gradient update*

$$\mathbf{w}_{t+1} = \mathbf{w}_t - \eta_t \left( n \nabla f(\mathbf{w}_t, \mathcal{D}) + \mathbf{z}_\tau \right), \qquad \mathbf{z}_\tau \sim \mathcal{N}(0, \tau^2 \sigma^2 \mathbf{I}),$$

*where $\tau \in (0, 1]$ is the ratio of $\sigma$ in the Gaussian noise. Let $\mathbf{w}_T^{(\tau)}$ be the $T$-th iterate produced with noise scale $\tau$ and $\mathbf{w}_T$ is without scaling.*

We follow the utility guarantee of the gradient methods in strong convex case [9].

*Proof.* Define

$$\mathcal{M}_t = n \nabla f(\mathbf{w}_t, z_t) + \mathbf{z}_t, \qquad \mathcal{M}_t^{(\tau)} = n \nabla f(\mathbf{w}_t, z_t) + \mathbf{z}_t^{(\tau)},$$

where $z_t$ is sampled uniformly from the dataset, $\mathbf{z}_t \sim \mathcal{N}(\mathbf{0}, \sigma^2 \mathbf{I})$, and $\mathbf{z}_t^{(\tau)} \sim \mathcal{N}(\mathbf{0}, \tau^2 \sigma^2 \mathbf{I})$.

By taking expectation over $z_t$ and then over the noise, we can obtain

$$\mathbb{E}[\mathcal{M}_t \mid \mathbf{w}_t] = n \cdot \frac{1}{n}\sum_{i=1}^{n} \nabla f(\mathbf{w}_t, z_i) + \mathbb{E}[\mathbf{z}_t] = \nabla F(\mathbf{w}_t),$$

and similarly $\mathbb{E}[\mathcal{M}_t^{(\tau)} \mid \mathbf{w}_t] = \nabla F(\mathbf{w}_t)$, which indicates both are unbiased estimators for gradient. Moreover, since each $f(\cdot, z)$ is $L$-Lipschitz and the noise is independent of the gradient,

$$\mathbb{E}\|\mathcal{M}_t\|_2^2 = \mathbb{E}\big\|n\nabla f(\mathbf{w}_t, z_t)\big\|_2^2 + 2\,\mathbb{E}\langle n\nabla f(\mathbf{w}_t, z_t), \mathbf{z}_t\rangle + \mathbb{E}\|\mathbf{z}_t\|_2^2$$

$$\leq n^2 L^2 + 0 + d\,\sigma^2 =: G^2, \quad \mathbb{E}\|\mathcal{M}_t^{(\tau)}\|_2^2 \leq n^2 L^2 + d\,\tau^2\sigma^2 =: G_\tau^2.$$

By Lemma 3 with $\eta_t = 1/(\lambda t)$,

$$\mathbb{E}\big[F(\mathbf{w}_T) - F(\tilde{\mathbf{w}})\big] = \mathcal{O}\Big(\tfrac{G^2 \log T}{\lambda T}\Big), \quad \mathbb{E}\big[F(\mathbf{w}_T^{(\tau)}) - F(\tilde{\mathbf{w}})\big] = \mathcal{O}\Big(\tfrac{G_\tau^2 \log T}{\lambda T}\Big).$$

Subtracting gives

$$\mathbb{E}\big[F(\mathbf{w}_T) - F(\mathbf{w}_T^{(\tau)})\big] = \mathcal{O}\Big(\tfrac{(G^2 - G_\tau^2)\log T}{\lambda T}\Big) = \mathcal{O}\Big(\tfrac{d\,\sigma^2(1-\tau^2)\log T}{\lambda T}\Big).$$

Similarly, for constant step size $\eta = c/\lambda$ ($0 < c \leq \frac{1}{2}$), Lemma 3 yields

$$\mathbb{E}\big[F(\mathbf{w}_T) - F(\tilde{\mathbf{w}})\big] = \mathcal{O}(\eta\,G^2), \quad \mathbb{E}\big[F(\mathbf{w}_T^{(\tau)}) - F(\tilde{\mathbf{w}})\big] = \mathcal{O}(\eta\,G_\tau^2),$$

hence

$$\mathbb{E}\big[F(\mathbf{w}_T) - F(\mathbf{w}_T^{(\tau)})\big] = \mathcal{O}\big(\eta\,(G^2 - G_\tau^2)\big) = \mathcal{O}\big(c\,d\,\sigma^2(1-\tau^2)\big).$$

$\square$

## C   Experiments

### C.1   Experimental Settings

We provide the dataset statistics in Table 6, including sample size, dimensionality, and number of classes for each dataset.

Table 6: Summary of benchmark datasets used in the experiments.

| Dataset | # samples ($n$) | dims ($d$) | classes ($c$) |
|---|---|---|---|
| Cornell | 827 | 4,134 | 7 |
| HHAR | 10,229 | 561 | 6 |
| USPS | 9,298 | 256 | 10 |
| ISOLET | 1,560 | 617 | 26 |
| Dermatology | 366 | 34 | 6 |
| Vehicle | 946 | 18 | 4 |

We present the experimental details of the DP SVMs: weight-perturbation settings are summarised in Table 7, and gradient-perturbation settings in Table 8. For weight perturbation, the regularization constant $C/n$ is fixed across methods, as it governs the standard deviation of the Gaussian noise; the search space is $\{0.001, 0.005, 0.01, 0.05, 0.10, 1.0\}$. For gradient perturbation, we adopt the base learning rate (Base LR) and regularization $C/n$ provided in the official implementation of Nie et al. [17]. Each method is fine-tuned over epochs $\{5, 10, 20, 30\}$ and learning-rate scales $\{0.1, 0.5, 1.0, 2.0, 5.0\}$; the resulting learning rate is Base LR $\times$ LR scale. Hyperparameters are selected at $\epsilon = 4$ and used for all other privacy budgets.

### C.2   Additional Experiments

We present additional results on the accuracy gap shown in Fig. 2 for the remaining datasets. Fig. 4 reports the results for weight-perturbation methods, and Fig. 5 shows the results for gradient-perturbation methods. We present additional results on the convergence shown in Fig. 3 for the remaining datasets in Fig. 6.

Table 7: Regularization constant $\frac{C}{n}$ used in all the weight-perturbation methods.

| Dataset | Cornell | Dermatology | HHAR | ISOLET | USPS | Vehicle |
|---|---|---|---|---|---|---|
| $\frac{C}{n}$ | 0.005 | 0.005 | 0.001 | 0.001 | 0.005 | 0.001 |

Table 8: Search space and best hyperparameters of gradient perturbation methods.

| Dataset | Base LR | $\frac{C}{n}$ | GRPUA | | Linear | | PMSVM-GP | | PMSVM-AGP | |
|---|---|---|---|---|---|---|---|---|---|---|
| | | | Epochs | LR Scale | Epochs | LR Scale | Epochs | LR Scale | Epochs | LR Scale |
| Cornell | 0.10 | 0.005 | 5 | 2.0 | 10 | 0.1 | 10 | 0.1 | 30 | 0.1 |
| HHAR | 0.02 | 0.0005 | 30 | 0.5 | 20 | 0.5 | 30 | 0.1 | 30 | 0.1 |
| USPS | 0.01 | 0.001 | 30 | 5.0 | 30 | 0.5 | 20 | 0.5 | 20 | 0.5 |
| ISOLET | 0.001 | 0.001 | 20 | 1.0 | 30 | 2.0 | 30 | 2.0 | 30 | 5.0 |
| Dermatology | 0.01 | 0.100 | 5 | 1.0 | 10 | 0.5 | 10 | 0.5 | 10 | 2.0 |
| Vehicle | 0.05 | 0.0001 | 20 | 1.0 | 10 | 0.5 | 10 | 1.0 | 30 | 1.0 |

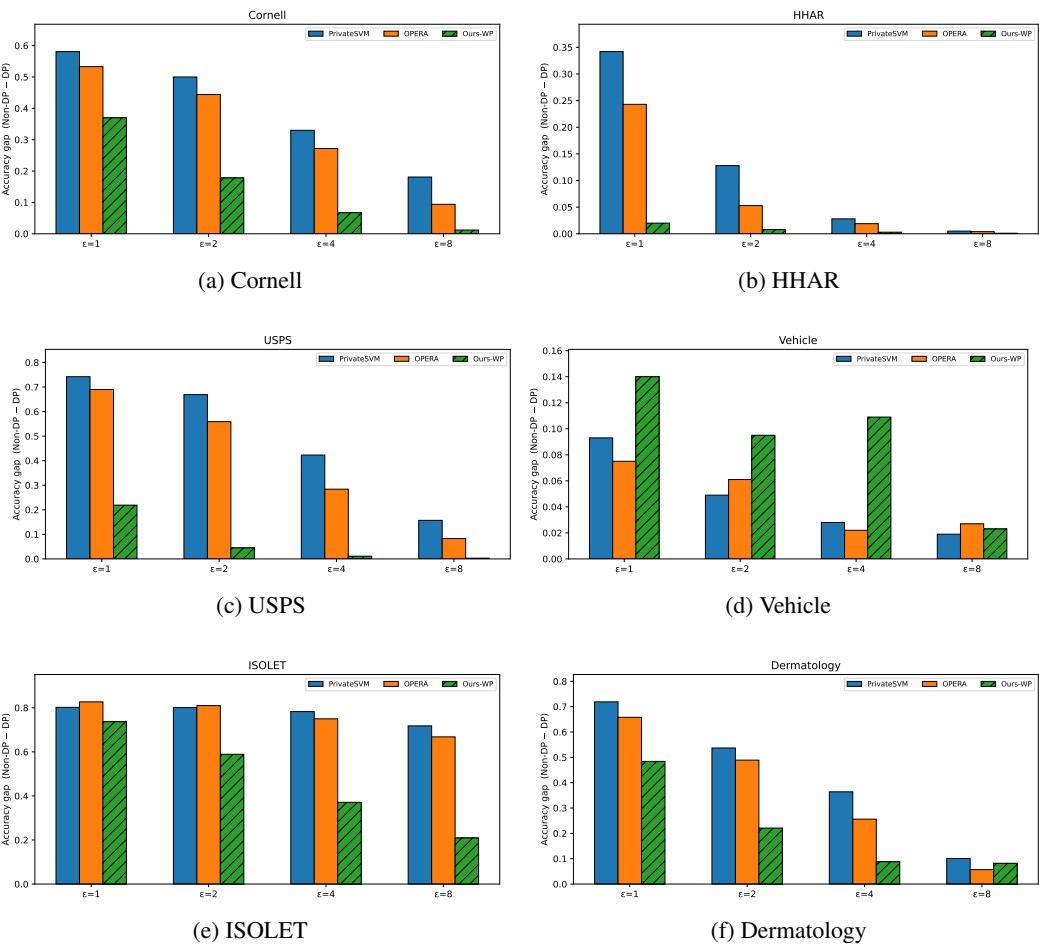

(a) Cornell   (b) HHAR

(c) USPS   (d) Vehicle

(e) ISOLET   (f) Dermatology

Figure 4: Weight Perturbation; Accuracy gap between DP-SVM methods and their non-private baselines ($\epsilon = \infty$). Lower value indicates a smaller accuracy–privacy trade-off, thus indicating a DP-friendly property.

We present additional results on the ablation studies of learning rate shown in Table 3 for the remaining datasets in Table 9. Furthermore, Table 10 shows the difference between selecting $R$. It is true that there is no universal rule to choose $R$, we concluded that $R = 1$ from [31] is a reasonable choice for the gradient method [33]. Table 11 shows the results of different batch sizes for the Poisson

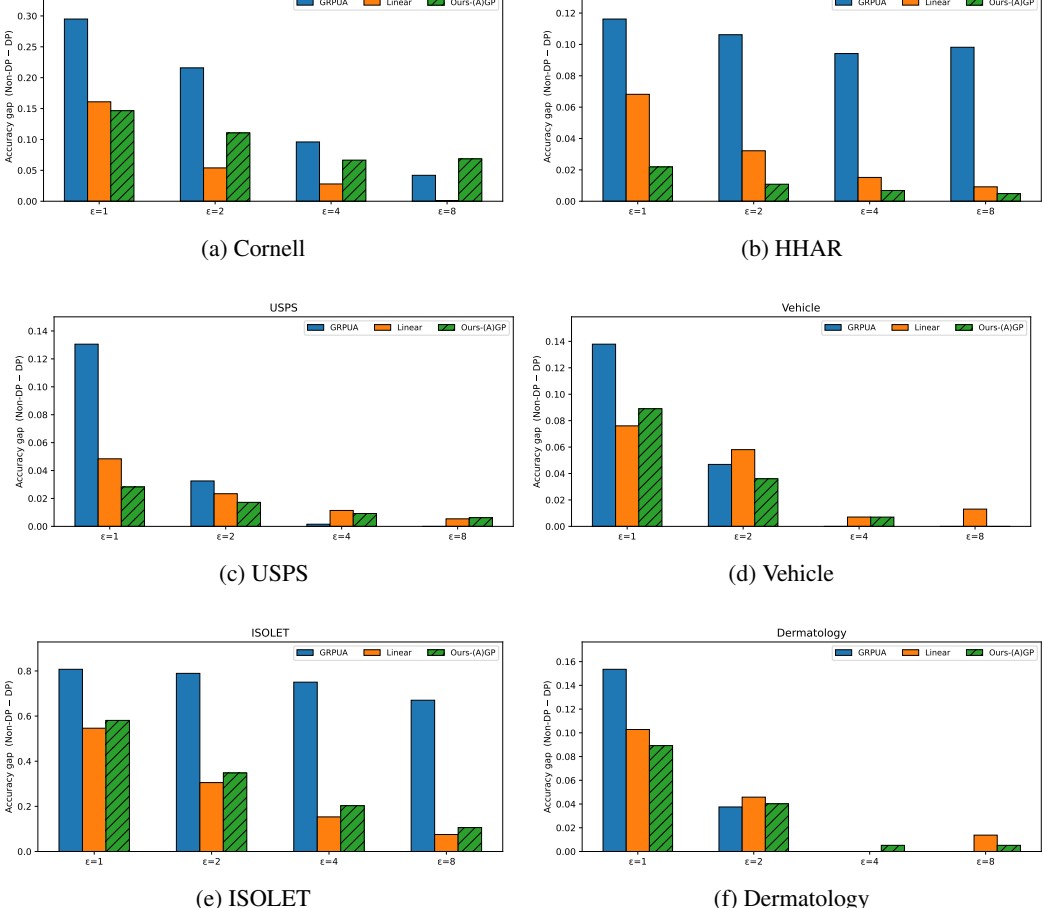

Figure 5: Gradient Perturbation; Accuracy gap between DP-SVM methods and their non-private baselines ($\epsilon = \infty$). Lower value indicates a smaller accuracy–privacy trade-off, thus indicating a DP-friendly property.

subsampling in Opacus [29]. We use min(batch size, # of training data). Compared to full batch gradients, the results of subsampling show better performance.

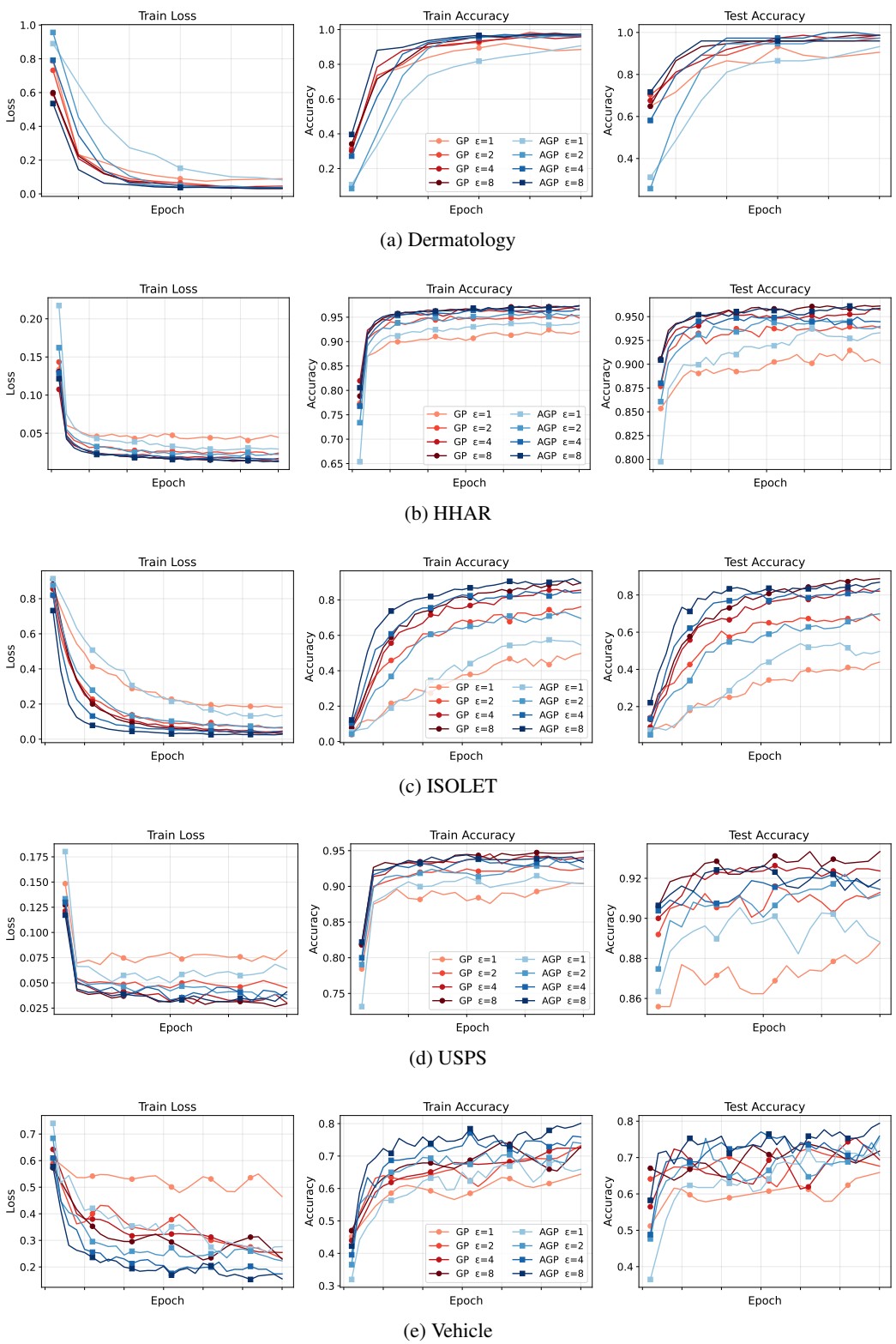

Figure 6: Convergence curves of training loss, training accuracy, and test accuracy for the proposed PMSVM-GP and PMSVM-AGP methods.

Table 9: Ablation study on the effect of learning rate decay for the proposed gradient perturbation methods. We bold the better performance in **bold** and Diff indicates the absolute difference w/ and w/o lr decay.

| Dataset | $\epsilon$ | PMSVM-GP | + lr decay | Diff | PMSVM-AGP | + lr decay | Diff |
|---|---|---|---|---|---|---|---|
| Cornell | 1 | 0.663±0.010 | **0.673**±0.033 | 0.010 | **0.692**±0.025 | **0.692**±0.022 | 0.000 |
| | 2 | 0.719±0.023 | **0.743**±0.009 | 0.024 | **0.728**±0.018 | 0.706±0.021 | 0.022 |
| | 4 | **0.771**±0.014 | 0.752±0.010 | 0.019 | **0.772**±0.019 | 0.748±0.013 | 0.024 |
| | 8 | **0.770**±0.012 | 0.765±0.015 | 0.005 | 0.769±0.029 | **0.774**±0.014 | 0.005 |
| Dermatology | 1 | 0.895±0.038 | **0.908**±0.029 | 0.013 | **0.900**±0.031 | 0.824±0.065 | 0.076 |
| | 2 | **0.949**±0.022 | 0.938±0.043 | 0.011 | **0.941**±0.024 | 0.930±0.040 | 0.011 |
| | 4 | **0.973**±0.017 | 0.938±0.021 | 0.035 | **0.984**±0.006 | 0.973±0.010 | 0.011 |
| | 8 | **0.976**±0.015 | 0.957±0.026 | 0.019 | **0.984**±0.006 | **0.984**±0.006 | 0.000 |
| HHAR | 1 | 0.938±0.008 | **0.942**±0.003 | 0.004 | **0.945**±0.002 | 0.941±0.001 | 0.004 |
| | 2 | **0.953**±0.004 | **0.953**±0.002 | 0.000 | **0.956**±0.001 | 0.949±0.003 | 0.007 |
| | 4 | **0.960**±0.003 | 0.956±0.002 | 0.004 | **0.959**±0.004 | 0.956±0.003 | 0.003 |
| | 8 | **0.962**±0.002 | 0.956±0.002 | 0.006 | 0.959±0.003 | **0.959**±0.003 | 0.000 |
| ISOLET | 1 | **0.336**±0.027 | 0.312±0.023 | 0.024 | **0.339**±0.033 | 0.246±0.021 | 0.093 |
| | 2 | **0.542**±0.054 | 0.496±0.031 | 0.046 | **0.572**±0.022 | 0.497±0.043 | 0.075 |
| | 4 | **0.717**±0.047 | 0.645±0.047 | 0.072 | **0.714**±0.045 | 0.667±0.038 | 0.047 |
| | 8 | **0.789**±0.017 | 0.687±0.022 | 0.102 | **0.814**±0.023 | 0.789±0.021 | 0.025 |
| USPS | 1 | 0.896±0.007 | **0.908**±0.004 | 0.012 | 0.908±0.005 | **0.912**±0.005 | 0.004 |
| | 2 | 0.917±0.005 | **0.922**±0.001 | 0.005 | 0.919±0.004 | **0.921**±0.001 | 0.002 |
| | 4 | **0.927**±0.004 | 0.924±0.003 | 0.003 | **0.926**±0.003 | 0.926±0.003 | 0.000 |
| | 8 | **0.930**±0.002 | 0.927±0.002 | 0.003 | 0.927±0.004 | **0.929**±0.003 | 0.002 |
| Vehicle | 1 | 0.604±0.083 | **0.666**±0.030 | 0.062 | **0.671**±0.029 | 0.662±0.021 | 0.009 |
| | 2 | 0.678±0.033 | **0.709**±0.027 | 0.031 | **0.724**±0.016 | 0.707±0.025 | 0.017 |
| | 4 | 0.727±0.030 | **0.729**±0.020 | 0.002 | **0.753**±0.013 | 0.739±0.011 | 0.014 |
| | 8 | 0.741±0.028 | **0.749**±0.016 | 0.008 | **0.768**±0.015 | 0.760±0.011 | 0.008 |

Table 10: Ablation study on the effect of selecting $R \in \{0.01, 0.1, 1, 10\}$.

| Dataset | $\epsilon$ | $R = 0.01$ | $R = 0.1$ | $R = 1$ | $R = 10$ |
|---|---|---|---|---|---|
| Cornell | 1 | 0.677 ± 0.007 | 0.681 ± 0.000 | 0.693 ± 0.032 | 0.347 ± 0.110 |
| | 2 | 0.679 ± 0.003 | 0.687 ± 0.006 | 0.707 ± 0.023 | 0.560 ± 0.034 |
| | 4 | 0.683 ± 0.003 | 0.745 ± 0.017 | 0.752 ± 0.023 | 0.653 ± 0.025 |
| | 8 | 0.747 ± 0.006 | 0.765 ± 0.024 | 0.765 ± 0.024 | 0.657 ± 0.006 |
| Dermatology | 1 | 0.842 ± 0.028 | 0.878 ± 0.070 | 0.905 ± 0.017 | 0.171 ± 0.110 |
| | 2 | 0.950 ± 0.016 | 0.955 ± 0.028 | 0.951 ± 0.042 | 0.230 ± 0.084 |
| | 4 | 0.987 ± 0.000 | 0.978 ± 0.016 | 0.978 ± 0.012 | 0.559 ± 0.034 |
| | 8 | 0.982 ± 0.008 | 0.978 ± 0.016 | 0.976 ± 0.018 | 0.743 ± 0.036 |
| HHAR | 1 | 0.922 ± 0.004 | 0.929 ± 0.002 | 0.929 ± 0.007 | 0.885 ± 0.007 |
| | 2 | 0.943 ± 0.001 | 0.947 ± 0.002 | 0.946 ± 0.004 | 0.913 ± 0.004 |
| | 4 | 0.948 ± 0.001 | 0.951 ± 0.002 | 0.956 ± 0.006 | 0.931 ± 0.002 |
| | 8 | 0.953 ± 0.001 | 0.953 ± 0.001 | 0.959 ± 0.003 | 0.938 ± 0.003 |
| ISOLET | 1 | 0.431 ± 0.007 | 0.458 ± 0.013 | 0.501 ± 0.025 | 0.057 ± 0.030 |
| | 2 | 0.661 ± 0.027 | 0.662 ± 0.026 | 0.687 ± 0.017 | 0.076 ± 0.029 |
| | 4 | 0.732 ± 0.013 | 0.746 ± 0.010 | 0.804 ± 0.010 | 0.119 ± 0.022 |
| | 8 | 0.825 ± 0.024 | 0.849 ± 0.014 | 0.840 ± 0.013 | 0.110 ± 0.002 |
| USPS | 1 | 0.917 ± 0.003 | 0.920 ± 0.001 | 0.897 ± 0.006 | 0.810 ± 0.004 |
| | 2 | 0.922 ± 0.001 | 0.927 ± 0.000 | 0.907 ± 0.006 | 0.856 ± 0.003 |
| | 4 | 0.927 ± 0.002 | 0.929 ± 0.003 | 0.917 ± 0.002 | 0.873 ± 0.002 |
| | 8 | 0.928 ± 0.002 | 0.931 ± 0.002 | 0.924 ± 0.003 | 0.891 ± 0.003 |
| Vehicle | 1 | 0.641 ± 0.026 | 0.680 ± 0.050 | 0.696 ± 0.060 | 0.329 ± 0.010 |
| | 2 | 0.684 ± 0.017 | 0.716 ± 0.009 | 0.753 ± 0.007 | 0.484 ± 0.019 |
| | 4 | 0.710 ± 0.015 | 0.727 ± 0.014 | 0.733 ± 0.023 | 0.578 ± 0.071 |
| | 8 | 0.722 ± 0.009 | 0.741 ± 0.020 | 0.766 ± 0.009 | 0.673 ± 0.038 |

Table 11: Ablation study on the effect of batch size for subsampling.

| Dataset | $\epsilon$ | bs=32 | bs=64 | bs=128 | bs=256 | bs=512 | bs=full |
|---|---|---|---|---|---|---|---|
| Cornell | 1 | $0.478 \pm 0.112$ | $0.681 \pm 0.000$ | $0.693 \pm 0.032$ | $0.408 \pm 0.130$ | $0.424 \pm 0.041$ | $0.480 \pm 0.066$ |
| | 2 | $0.564 \pm 0.021$ | $0.687 \pm 0.000$ | $0.707 \pm 0.023$ | $0.584 \pm 0.034$ | $0.544 \pm 0.019$ | $0.590 \pm 0.024$ |
| | 4 | $0.641 \pm 0.058$ | $0.737 \pm 0.015$ | $0.752 \pm 0.023$ | $0.630 \pm 0.021$ | $0.602 \pm 0.016$ | $0.645 \pm 0.034$ |
| | 8 | $0.667 \pm 0.019$ | $0.761 \pm 0.004$ | $0.765 \pm 0.024$ | $0.663 \pm 0.021$ | $0.661 \pm 0.023$ | $0.671 \pm 0.007$ |
| Dermatology | 1 | $0.297 \pm 0.059$ | $0.883 \pm 0.077$ | $0.905 \pm 0.017$ | $0.365 \pm 0.068$ | $0.260 \pm 0.227$ | $0.243 \pm 0.143$ |
| | 2 | $0.369 \pm 0.068$ | $0.937 \pm 0.034$ | $0.951 \pm 0.042$ | $0.379 \pm 0.116$ | $0.357 \pm 0.097$ | $0.320 \pm 0.021$ |
| | 4 | $0.599 \pm 0.056$ | $0.919 \pm 0.023$ | $0.978 \pm 0.012$ | $0.527 \pm 0.115$ | $0.522 \pm 0.123$ | $0.541 \pm 0.166$ |
| | 8 | $0.712 \pm 0.067$ | $0.964 \pm 0.028$ | $0.976 \pm 0.018$ | $0.680 \pm 0.090$ | $0.635 \pm 0.023$ | $0.635 \pm 0.059$ |
| HHAR | 1 | $0.878 \pm 0.009$ | $0.934 \pm 0.003$ | $0.929 \pm 0.007$ | $0.888 \pm 0.004$ | $0.886 \pm 0.012$ | $0.884 \pm 0.005$ |
| | 2 | $0.908 \pm 0.002$ | $0.941 \pm 0.004$ | $0.946 \pm 0.004$ | $0.916 \pm 0.006$ | $0.912 \pm 0.006$ | $0.913 \pm 0.008$ |
| | 4 | $0.928 \pm 0.001$ | $0.954 \pm 0.004$ | $0.956 \pm 0.006$ | $0.932 \pm 0.003$ | $0.928 \pm 0.004$ | $0.932 \pm 0.005$ |
| | 8 | $0.940 \pm 0.000$ | $0.950 \pm 0.002$ | $0.959 \pm 0.003$ | $0.939 \pm 0.004$ | $0.944 \pm 0.003$ | $0.943 \pm 0.003$ |
| ISOLET | 1 | $0.059 \pm 0.010$ | $0.465 \pm 0.042$ | $0.501 \pm 0.025$ | $0.063 \pm 0.008$ | $0.038 \pm 0.013$ | $0.037 \pm 0.014$ |
| | 2 | $0.054 \pm 0.022$ | $0.614 \pm 0.013$ | $0.687 \pm 0.017$ | $0.074 \pm 0.031$ | $0.060 \pm 0.038$ | $0.037 \pm 0.005$ |
| | 4 | $0.124 \pm 0.034$ | $0.769 \pm 0.039$ | $0.804 \pm 0.010$ | $0.093 \pm 0.034$ | $0.084 \pm 0.012$ | $0.105 \pm 0.030$ |
| | 8 | $0.120 \pm 0.013$ | $0.834 \pm 0.018$ | $0.840 \pm 0.013$ | $0.157 \pm 0.010$ | $0.145 \pm 0.006$ | $0.144 \pm 0.045$ |
| USPS | 1 | $0.813 \pm 0.002$ | $0.918 \pm 0.002$ | $0.897 \pm 0.006$ | $0.803 \pm 0.007$ | $0.813 \pm 0.012$ | $0.829 \pm 0.008$ |
| | 2 | $0.855 \pm 0.011$ | $0.922 \pm 0.001$ | $0.907 \pm 0.006$ | $0.854 \pm 0.006$ | $0.854 \pm 0.014$ | $0.854 \pm 0.009$ |
| | 4 | $0.876 \pm 0.004$ | $0.920 \pm 0.003$ | $0.917 \pm 0.002$ | $0.880 \pm 0.004$ | $0.878 \pm 0.001$ | $0.874 \pm 0.005$ |
| | 8 | $0.891 \pm 0.002$ | $0.931 \pm 0.003$ | $0.924 \pm 0.003$ | $0.899 \pm 0.006$ | $0.895 \pm 0.003$ | $0.887 \pm 0.010$ |
| Vehicle | 1 | $0.363 \pm 0.163$ | $0.688 \pm 0.024$ | $0.696 \pm 0.060$ | $0.455 \pm 0.063$ | $0.444 \pm 0.032$ | $0.424 \pm 0.064$ |
| | 2 | $0.410 \pm 0.094$ | $0.684 \pm 0.024$ | $0.753 \pm 0.007$ | $0.480 \pm 0.063$ | $0.445 \pm 0.131$ | $0.480 \pm 0.075$ |
| | 4 | $0.524 \pm 0.080$ | $0.718 \pm 0.018$ | $0.733 \pm 0.023$ | $0.569 \pm 0.034$ | $0.563 \pm 0.051$ | $0.571 \pm 0.031$ |
| | 8 | $0.651 \pm 0.056$ | $0.731 \pm 0.007$ | $0.766 \pm 0.009$ | $0.659 \pm 0.020$ | $0.624 \pm 0.031$ | $0.639 \pm 0.040$ |

