# OpenReview forum: "Multi-Class Support Vector Machine with Differential Privacy"
_NeurIPS.cc/2025/Conference — NeurIPS 2025 poster_

### Official Review · Reviewer_EHLs · 2025-06-23

**Clarity:** 4
**Significance:** 3
**Originality:** 3
**Rating:** 4
**Confidence:** 4

**Summary:**

This paper presents a differentially private multi-class Support Vector Machine (PMSVM) designed to enhance data privacy in machine learning. It addresses the inefficiency of traditional multi-class SVM approaches (one-versus-rest and one-versus-one), which repeatedly access data, by proposing all-in-one SVM methods. These methods reduce privacy budget consumption by requiring only a single access to each data sample when constructing multi-class SVM boundaries. The paper details two primary methods for achieving privacy: Weight Perturbation (WP) and Gradient Perturbation (GP), providing rigorous analysis for both. Empirical results presented demonstrate that this PMSVM approach outperforms existing differentially private SVM methods in multi-class classification scenarios in most cases.

**Questions:**

1. See weaknesses
2. The differential privacy of the proposed work is only established theoretically. Is it possible to empirically evaluate the differential privacy of the proposed methods compared to the all-in-one SVMs [12-15], perhaps with a membership inference attack?
3. SVM models are already very lightweight and thus fast compared to deep learning models. What is the realistic impact of the proposed method to be more efficient, particularly for the Gradient Perturbation-based approach? In particular, what type of scenario or application would benefit from the proposed method, since the total time difference seems small?

**Ethical Concerns:**

["NO or VERY MINOR ethics concerns only"]

**Final Justification:**

My points have been taken and my rating has been adjusted accordingly.

**Limitations:**

Yes

**Quality:**

3

**Strengths And Weaknesses:**

Strengths
- The paper is well written and presents clear arguments
- The work presents a theoretical analysis of the proposed method to demonstrate differential privacy and illustrate time complexity
- The proposed method seems to outperform the baselines in most cases

Weaknesses
- The paper is motivated by the inefficiency of applying differential privacy to SVM. However, the work skips some baselines, and is not more efficient than OPERA, for the runtime measurements (Table 4)
- The work has a limited scope of baseline comparison and deliberately excludes certain types of differentially private Support Vector Machine (DP-SVM) models from its comparisons.
- The work neglects to discuss some related works:
  - Privacy-preserving multi-class support vector machine model on medical diagnosis
  - A novel weighted support vector machines multiclass classifier based on differential evolution for intrusion detection systems

---

> ### Author Rebuttal · Authors · 2025-07-30
>
> First, we thank the reviewer for suggesting we find more related work and explore realistic scenarios. We now answer each question individually.
>
> **W1 Inefficiency of proposed method (and runtime measurements)**
>
> First, we make it clear that we do not use the term “efficiency” to explain our strength, which might confuse the reader about our strength in computational time. We will also revise the “efficiency” in line 114, even though it is not directly connected to this idea.
>
> **Inefficiency.** We agree that the reader might be confused about how the all-in-one SVM enhances the existing OvR approaches when applying differential privacy to SVM. However, our goal is to reduce the number of data accesses to individual data samples; thus, “all-in-one SVMs utilized in this work avoid redundant data accesses compared to existing one-vs-rest and one-vs-one approaches for multi-class DP-SVM, which reduce the redundant consumption for privacy budget,” noted by reviewer [**orwF**]. Although we know that the reviewer is already familiar with the dilemma of data access and privacy, we will explain it again for a clear understanding of the paper.
>
> The composition theorem, one of differential privacy's most fundamental properties, is presented in Remark 1 (line 65) for two mechanisms. When we consider the general composition for $(\epsilon,\delta)$‑d.p. algorithms for multiple $k\ge2$ mechanisms [1], we can similarly formulate the composition as follows (shorted version):
>
> ---
>
> Let $\mathcal{M_1}\colon D\to\mathcal{C_1}$ be $(\epsilon,\delta)$‑d.p., and for each $k\ge2$ let
> $\mathcal{M_k}\colon (D, s_{k-1},\dots,s_{1}) \to\mathcal{C_k}$
> be $(\epsilon,\delta)$‑d.p. for all given $(s_{k-1},\dots,s_{1})\in\bigotimes_{j=1}^{k-1}\mathcal{C_j}$.
> Then for any neighboring datasets $D,D'$ and any $S\subseteq\bigotimes_{j=1}^{k-1}\mathcal{C_j}$,
> $$\Pr\bigl((\mathcal{M_1},\dots,\mathcal{M_k})\in S\bigr)\le e^{k\epsilon}\Pr'\bigl((\mathcal{M_1},\dots,\mathcal{M_k})\in S\bigr)+k\delta$$
>
> ---
>
> Simply, the composition theorem indicates that the number of accesses to data $D$ directly affects the noise level if each mechanism has a dependency on training data. When applying this principle to the OvR case, the composition of c classifiers requires c$\epsilon$ privacy budget when each binary classifier requires $\epsilon$. Therefore, to maintain the same total privacy budget (e.g., $\epsilon$=8), we can only allocate $\epsilon=\frac{8}{c}$ to each binary classifier, amplifying the amount of noise on each classifier. Note that DPSGD has similar constraints on training steps despite utilizing modern composition theorems such as batch sampling.
>
> In contrast, our method PMSVM requires only one data access to build a multi-class classifier, allowing us to utilize the full privacy budget $\epsilon$. This is why we propose a multi-class DP SVM based on the all-in-one method.
>
> [1] The algorithmic foundations of differential privacy, Foundations and trends in theoretical computer science (2014)
>
> **Computational time.** To address the reviewer’s concern about runtime, we restate that we design two approaches, i.e., weight perturbation and gradient perturbation.
>
> Regarding runtime between OPERA and PMSVM‑WP, we use scikit‑learn’s “LinearSVC,” which implements the OvR method for multi‑class classification for OPERA. In contrast, PMSVM‑WP uses the “crammer_singer” option.
>
> Thus, the difference relies only on the implementation of two methods in sklearn because the proposed weight perturbation requires only an additional single noise‑addition step: $O(1)$ per parameter, i.e., $O(\text{num of params})$ overall.
>
> To explain the computational part, we can think that the Crammer–Singer formulation solves a single joint QP with $nc$ variables, whereas OvR decomposes into $c$ independent binary SVMs, each with $n$ variables. Given that standard QP solvers scale as $O(\text{num of params}^3)$, Crammer–Singer entails $O(n^3c^3)$, while OvR requires $O(cn^3)$. In practice (e.g., in scikit‑learn using LIBLINEAR), the observed gap is about $O(c)$ times practically. This explains the runtime gap between OPERA and PMSVM‑WP. We will add this discussion to the main paper and the appendix on weight‑perturbation computation.
>
> On the other hand, for gradient perturbation, because the model is updated in a single pass rather than via sequential OvR training, PMSVM‑GP is superior to GRPUA in both performance and computational efficiency.
>
> **W2 Limited Scope and Comparison Method**
>
> We agree that the readers can think that our paper may have a limited scope of comparison methods. Nonetheless, the methods included represent the principal prior DP techniques for SVM. In summary, we introduce various DP-SVM papers as summarized in the table below.
>
> | Algorithm | Privacy | Noise Injection | Model Publish | Multi Class |
> | --- | --- | --- | --- | --- |
> | **PrivateSVM (Rubinstein et al., 2012)** | $(\epsilon, 0)$ | Weight | O | X |
> | DPKL‑PINP (Jain et al., 2013) | $(\epsilon, \delta)$ | Output | X | X |
> | **OPERA (Ding et al., 2022)** | $(\epsilon, \delta)$ | Weight | O | X |
> | **GRPUA (Ding et al., 2022)** | $(\epsilon, \delta)$ | Gradient | O | X |
> | Ours | $(\epsilon, \delta)$ | Weight (WP) / Gradient (GP) | O | O |
>
> The only related work we omitted from the comparison is DPKL (Jain et al., 2013), which requires interaction with end users and therefore cannot publicly release the trained SVM’s hyperparameters. Among the previous works, existing binary DP-SVMs cannot be directly extended to multi-class settings because OvO/OvR training reuses the dataset multiple times. Hence, despite the small pool, we believe our comparisons can show our own strength.
>
> We address additional related work in response to W3 below.
>
> **W3. More Related works**
>
> At the time of the submission, we initially excluded domain-specific approaches (e.g., medical or IoT) mentioned by the reviewer because we think our focus is on general DP algorithms. In rebuttal, we agreed with the reviewer's comment about investigating the domain-specific paper. We first took a look at the papers the reviewer suggested.
>
> [1] Privacy-preserving multi-class support vector machine model on medical diagnosis, IEEE Journal of Biomedical and Health Informatics (2022).
>
> - This paper proposes a cryptographic scheme for privacy-preserving multi-class SVM; this is orthogonal to DP but worth noting.
>
> [2] A novel weighted support vector machines multiclass classifier based on differential evolution for intrusion detection systems, Information Sciences (2017)
>
> - This paper employs differential evolution rather than differential privacy and is thus out of scope.
>
> Following the reviewers' suggestion, we found some more papers that can be relevant in terms of SVM in DP versions, or their application in specific domains.
>
> [3] Z. Sun, J. Yang, X. Li, Differentially Private Singular Value Decomposition for Training Support Vector Machines, Computational Intelligence and Neuroscience (2022)
>
> - This paper investigates data-wise perturbation (called local DP) for SVM with singular value decomposition.
>
> [4] Differentially private image classification using support vector machine and differential privacy, Machine Learning and Knowledge Extraction (2019)
>
> [5] Privacy-preserving RBF kernel support vector machine, BioMed research international (2014)
>
> - These methods adopt PrivateSVM in their specific domain, such as medical settings.
>
> [6] Differentially Private Kernel Support Vector Machines Based on the Exponential and Laplace Hybrid Mechanism., Security and Communication Networks (2021)
>
> [7] A differential privacy support vector machine classifier based on dual variable perturbation, IEEE Access 7 (2019)
>
> - We additionally find more methods for DP-SVMs, but perturbing only either dual variables or support vectors cannot guarantee the DP of outputs.
>
> We will incorporate these works as related works.
>
> **Q3. Realistic impact of the proposed method in terms of efficiency or application**
>
> As discussed in **W1**, our primary contribution is improved generalization of DP‑SVMs in multi‑class settings, rather than computational efficiency. Because SVMs are compact, they remain useful compared with deep learning, as the reviewer noted. In practice, multi‑class classification is more common than binary classification. Prior DP‑SVM work largely targets binary setups, which limits practical impact. In contrast, our method provides a DP model that is (i) a lightweight, fast SVM relative to deep learning models and (ii) naturally suited to multi‑class scenarios.
>
> In terms of applications, our approach aligns with the above domain-specific papers, including medical data and resource‑constrained devices such as IoT.
>
> **[General]** Additionally, we kindly request the reviewer to take a look at any interesting topics during the rebuttal period. In summary, we received positive feedback on: (i) reducing redundant privacy budget consumption for multi-class SVM in DP scenarios [Fj1B, orwF, hEqw, EHLs], (ii) empirical strengths [N89T, Fj1B, hEqw, EHLs], and (iii) well-written and well-executed paper [Fj1B, hEqw].
>
> For each rebuttal concerning “weakness” and “question”, we addressed the following topics:
>
> - [**N89T**] Our novelty and the importance of single data access in DP, and details of theorems
> - [**Fj1B**] Clarity of Lemma 1, convexity analysis of Eq. (12), and selection of hyperparameter $R$
> - [**orwF**] DPSVM in LLM era, selection of all-in-one base model, and advanced techniques in DP-SGD
> - [**hEqw**] Sub-sampling in gradient descent, low performance, and limitations
> - [**EHLs**] Inefficiency, additional comparison methods/related works, membership inference attack, and realistic impacts
>
> If the reviewer has any questions about these topics, please refer to our rebuttals for the corresponding reviewers.
>
> **We will continue the remaining response of **Q2. Empirical evaluation of membership inference attack** in the comments below.**

---

> > ### Author Response · Authors · 2025-08-01
> >
> > **Q2. Membership inference attack compared to non-DP All-in-one SVM**
> >
> > As the reviewer suggested, we evaluated the robustness of our private DP-SVM method against a membership inference attack (MIA). We selected the simplest attack, LOSS (using loss with a global threshold) [1], which computes a threshold from the training loss and classifies samples as “member” or “non-member.”
> >
> > Following the reviewer’s comment, we compared Ours-GP with $\epsilon=1,2$ and M$^3$-SVM  for non-DP all-in-one SVM. If the loss distributions for member and non-member samples are easily distinguishable, the attack achieves high recall and F1 scores. In MIA, recall means the portion of sensitive member data that is detected as members, and F1 indicates the overall performance of binary classification. To implement the MIA, we split the data into member (training) and non-member (test) sets, trained only on the member data, and then applied a threshold-based classifier at inference.
> >
> > We report average member loss, non-member loss, and loss ratio $\frac{\text{non-member loss}}{\text{member loss}}$ for MIA with LOSS [1]. A ratio near 1 indicates that samples are hard to distinguish; larger ratios mean easier detection, since the model yields lower loss on (over-)fitted member samples and higher loss on unseen non-member samples.  In particular, non-DP M$^3$-SVM achieves a recall close to 1.0 and a higher F1 score, indicating its vulnerability to MIA.
> >
> > | MIA | Method | $\epsilon$ | Member loss | Non-member loss | Loss ratio | Recall ($\downarrow$) | F1 Score ($\downarrow$) |
> > | --- | --- | --- | --- | --- | --- | --- | --- |
> > | Cornell | Ours | 1 | 34.067 | 41.897 | 1.230 | 0.766±0.017 | 0.621±0.009 |
> > |  | Ours | 2 | 23.396 | 40.622 | 1.736 | 0.806±0.010 | 0.640±0.007 |
> > |  | M$^3$-SVM | $\infty$ | 0.001 | 35.863 | >1000 | 0.992±0.003 | 0.729±0.010 |
> > | Dermatology | Ours | 1 | 5.417 | 8.668 | 1.600 | 0.851±0.030 | 0.642±0.015 |
> > |  | Ours | 2 | 3.675 | 7.738 | 2.106 | 0.908±0.020 | 0.658±0.008 |
> > |  | M$^3$-SVM | $\infty$ | 0.001 | 1.063 | >1000 | 0.981±0.009 | 0.671±0.007 |
> > | HHAR | Ours | 1 | 7.564 | 8.385 | 1.109 | 0.874±0.009 | 0.639±0.003 |
> > |  | Ours | 2 | 3.184 | 4.109 | 1.291 | 0.903±0.007 | 0.647±0.003 |
> > |  | M$^3$-SVM | $\infty$ | 0.349 | 1.084 | 3.107 | 0.975±0.016 | 0.664±0.004 |
> > | ISOLET | Ours | 1 | 37.029 | 42.001 | 1.134 | 0.589±0.021 | 0.560±0.018 |
> > |  | Ours | 2 | 24.697 | 30.579 | 1.238 | 0.659±0.013 | 0.595±0.015 |
> > |  | M$^3$-SVM | $\infty$ | 0.001 | 4.621 | >1000 | 0.990±0.004 | 0.692±0.005 |
> > | USPS | Ours | 1 | 11.566 | 12.863 | 1.112 | 0.867±0.013 | 0.637±0.006 |
> > |  | Ours | 2 | 9.372 | 12.278 | 1.310 | 0.886±0.011 | 0.642±0.005 |
> > |  | M$^3$-SVM | $\infty$ | 2.386 | 8.901 | 3.731 | 0.951±0.008 | 0.663±0.003 |
> > | Vehicle | Ours | 1 | 26.453 | 31.368 | 1.186 | 0.722±0.031 | 0.597±0.012 |
> > |  | Ours | 2 | 13.695 | 17.124 | 1.250 | 0.741±0.011 | 0.605±0.010 |
> > |  | M$^3$-SVM | $\infty$ | 2.821 | 3.690 | 1.308 | 0.793±0.039 | 0.623±0.023 |
> >
> > We conclude that our method not only reduces the privacy budget in DP-SVMs but also is more resistant to MIA than the non-DP all-in-one SVM. Note that attack performance could be further improved by tuning the threshold or using more advanced MIA techniques. We will include this result in the paper.
> >
> > [1] Privacy Risk in Machine Learning: Analyzing the Connection to Overfitting, *IEEE 31st computer security foundations symposium (CSF) (2018).*

---

> ### Author Response · Authors · 2025-08-04
> **Thank for comments**
>
> We are grateful for the reviewer’s careful and comprehensive comments on our rebuttal.
>
> Thanks to the detailed requests about related work and empirical evaluation, we believe that our work has significantly improved.

---

### Official Review · Reviewer_hEqw · 2025-06-23

**Clarity:** 3
**Significance:** 3
**Originality:** 2
**Rating:** 4
**Confidence:** 3

**Summary:**

Past work has considered differentially private versions of support vector machines, focusing on binary classification. This paper focuses on multi-class classification, arguing that methods based on reduction to many binary classifiers do not give a good privacy-utility trade-off as the number of classes grows. Instead, an "all in one" formulation that treats SVM multiclass classification as a single optimization problem, is used as a starting point. Empirical studies on standard benchmarks show that this approach most often outperform previous methods.

**Questions:**

- Did you consider privacy amplification by subsampling as a way of speeding up the algorithm?
- The classification accuracy is modest for most of the problems studied, even for rather large epsilon. Did you consider approximate DP and/or larger datasets for which it is easier to achieve high accuracy?

**Ethical Concerns:**

["NO or VERY MINOR ethics concerns only"]

**Final Justification:**

The authors provided a solid rebuttal including new results that improve the empirical part of the paper. My assessment remains positive.

**Limitations:**

The limitations discussed in the conclusion do not appear very thorough. For example, since the method is based on private gradient descent I would expect it to be slow for large datasets, which would be another limitation.

**Quality:**

4

**Strengths And Weaknesses:**

The paper is well-written and well carried out. The empirical results show that the proposed method is interesting in practice.

The main weakness is that the contribution is mainly empirical. The theoretical contribution essentially consists of putting together a known optimization problem and known techniques for private ERM. It would have been nice if techniques more modern than [3] had been used, for example amplification by subsampling which is key in analyzing private SGD (see e.g. https://arxiv.org/abs/2210.00597)

---

> ### Author Rebuttal · Authors · 2025-07-30
>
> First, we thank the reviewer for raising various theoretical questions and suggesting ways to improve experimental performance. We now answer the question individually.
>
> **W1, Q1 Sub-sampling techniques**
> We completely agree with the reviewer regarding the use of sub-sampling techniques. The paper suggested by the reviewer [1] provides mathematical guarantees for the composition method (which performs better than basic composition in Remark 1 or moment accounting in Lemma 2) for subsampling approaches. While this is not directly connected to the theoretical analysis in our paper, such as convergence (Theorem 3), we still utilize subsampling methods for improved composition.
>
> For the experiments, we first clarify that we used a batch size of 128 rather than full batch gradient descent. We apologize for this confusion and will clarify the use of subsampling methods in the paper. As the reviewer suggested, better subsampling methods can achieve improved performance. Fortunately, as noted on line 228, we use Opacus, a modern library developed by Meta Open Source. Opacus is actively updated with new DP-SGD algorithms, including optimizers and Poisson sampling strategies.
>
> The results using different batch sizes are shown in the table below. We can observe that Poisson sub-sampling techniques, as described in [1], demonstrate better performance than full batch gradient descent. However, when the batch size is too small, the noise effect becomes stronger, thus the performance decreases. This analysis aligned with deep learning with DP-SGD, which uses a comparably large batch size for calculating noisy gradients. With Poisson sampling, the batch size is not always guaranteed, but it indicates the approximate average.
>
> | Dataset | ε | bs$\\approx$32 | bs$\approx$64 | bs$\approx$128 | bs=full |
> | --- | --- | --- | --- | --- | --- |
> | Cornell | 1 | 0.478±0.112 | 0.681±0.000 | **0.693±0.032** | 0.480±0.066 |
> |  | 2 | 0.564±0.021 | 0.687±0.000 | **0.707±0.023** | 0.590±0.024 |
> |  | 4 | 0.641±0.058 | 0.737±0.015 | **0.752±0.023** | 0.645±0.034 |
> |  | 8 | 0.667±0.019 | 0.761±0.004 | **0.765±0.024** | 0.671±0.007 |
> | Dermatology | 1 | 0.297±0.059 | 0.883±0.077 | **0.905±0.017** | 0.243±0.143 |
> |  | 2 | 0.369±0.068 | 0.937±0.034 | **0.951±0.042** | 0.320±0.021 |
> |  | 4 | 0.599±0.056 | 0.919±0.023 | **0.978±0.012** | 0.541±0.166 |
> |  | 8 | 0.712±0.067 | 0.964±0.028 | **0.976±0.018** | 0.635±0.059 |
> | HHAR | 1 | 0.878±0.009 | **0.934±0.003** | 0.929±0.007 | 0.884±0.005 |
> |  | 2 | 0.908±0.002 | 0.941±0.004 | **0.946±0.004** | 0.913±0.008 |
> |  | 4 | 0.928±0.001 | 0.954±0.004 | **0.956±0.006** | 0.932±0.005 |
> |  | 8 | 0.940±0.000 | 0.950±0.002 | **0.959±0.003** | 0.943±0.003 |
> | ISOLET | 1 | 0.059±0.010 | 0.465±0.042 | **0.501±0.025** | 0.037±0.014 |
> |  | 2 | 0.054±0.022 | 0.614±0.013 | **0.687±0.017** | 0.037±0.005 |
> |  | 4 | 0.124±0.034 | 0.769±0.039 | **0.804±0.010** | 0.105±0.030 |
> |  | 8 | 0.120±0.013 | 0.834±0.018 | **0.840±0.013** | 0.144±0.045 |
> | USPS | 1 | 0.813±0.002 | **0.918±0.002** | 0.897±0.006 | 0.829±0.008 |
> |  | 2 | 0.855±0.011 | **0.922±0.001** | 0.907±0.006 | 0.854±0.009 |
> |  | 4 | 0.876±0.004 | **0.920±0.003** | 0.917±0.002 | 0.874±0.005 |
> |  | 8 | 0.891±0.002 | **0.931±0.003** | 0.924±0.003 | 0.887±0.010 |
> | Vehicle | 1 | 0.363±0.163 | 0.688±0.024 | **0.696±0.060** | 0.424±0.064 |
> |  | 2 | 0.410±0.094 | 0.684±0.024 | **0.753±0.007** | 0.480±0.075 |
> |  | 4 | 0.524±0.080 | 0.718±0.018 | **0.733±0.023** | 0.571±0.031 |
> |  | 8 | 0.651±0.056 | 0.731±0.007 | **0.766±0.009** | 0.639±0.040 |
>
> **Better Optimizer.** Since the development of DP-SGD in 2016, clipping-based gradient updates have been extensively studied, allowing us to leverage recent analytical techniques for gradient-based optimization. For the adaptive gradient perturbation in Equation 16, we employ the DP-Adam [2] optimizer, which is officially supported by the Opacus library. When we consider the private gradient of Theorem 2 as $\hat {\mathbf g}_ t$, the adaptive update formulation is as follows:
>
> $$\hat{\mathbf m} _ t = \beta_1 \hat{\mathbf m} _ {t-1} + (1-\beta_1)\hat{\mathbf g} _ t,\ \hat{\mathbf v} _ t = \beta_2 \hat{\mathbf v} _ {t-1} + (1-\beta_2)(\hat{\mathbf g} _ t \odot \hat{\mathbf g} _ t)$$
>
> $$\hat{\mathbf m} _ t = \frac{\hat{\mathbf m} _ t}{1-\beta_1^t},\ \hat{\mathbf v} _ t = \frac{\hat{\mathbf v} _ t}{1-\beta_2^t},\ \hat  {\mathbf w} _ t = \hat {\mathbf w} _ {t-1} - \eta\frac{\hat{\mathbf m} _ t}{\sqrt{\hat{\mathbf v} _ t} + \gamma}.$$
>
> Therefore, adopting advanced techniques is also feasible within our framework. We will include the details of the above analysis in the paper.
>
> [1] Composition of Differential Privacy & Privacy Amplification by Subsampling, Arxiv (2022)
>
> [2] Large Language Models Can Be Strong Differentially Private Learners, ICLR 2022
>
>
> **Q2. Low performance**
>
> We agree that overall performance remains low, despite our method outperforming previous DP-SVM methods. Although larger datasets might yield better performance, as the reviewer noted, our primary objective in choosing machine learning approaches is to address scenarios with limited data samples where deep learning models underperform.
>
> Instead, we show the difficulty of ensuring DP in small datasets, even for deep learning with DP. The linear model in Table 2 uses the same single nn.Linear architecture as ours, so we cannot demonstrate differences in "deep" properties. Therefore, we added experiments with "Residual," which employs a significantly deeper architecture with 9 linear transformation layers consisting of an input layer, four residual blocks (2 linear layers each), and an output head.
> The experimental results below indicate that DP deep learning methods also show poor performance on small datasets, and in some cases perform worse than simple Linear networks. Following [1], it is true for the small data case that deeper networks may not always be effective for DP methods. While we agree that better feature extractors [1] or alternative architectural designs [2] could be used for DP, we leave this as a future direction since our focus is on DPSVM.
>
>
> | Dataset | ε | Linear | **Residual (NEW)** | Ours |
> | --- | --- | --- | --- | --- |
> | Cornell | 1 | 0.624 | 0.685 | **0.693** |
> |  | 2 | 0.695 | 0.685 | **0.707** |
> |  | 4 | 0.747 | 0.699 | **0.752** |
> |  | 8 | **0.792** | 0.763 | 0.765 |
> | Dermatology | 1 | **0.911** | 0.752 | 0.905 |
> |  | 2 | 0.930 | 0.815 | **0.951** |
> |  | 4 | 0.970 | 0.901 | **0.978** |
> |  | 8 | 0.973 | 0.941 | **0.976** |
> | HHAR | 1 | 0.887 | **0.930** | 0.929 |
> |  | 2 | 0.920 | 0.940 | **0.946** |
> |  | 4 | 0.936 | 0.950 | **0.956** |
> |  | 8 | 0.949 | **0.960** | 0.959 |
> | ISOLET | 1 | 0.466 | 0.340 | **0.501** |
> |  | 2 | 0.672 | 0.571 | **0.687** |
> |  | 4 | **0.820** | 0.737 | 0.804 |
> |  | 8 | **0.858** | 0.827 | 0.840 |
> | USPS | 1 | 0.875 | 0.877 | **0.897** |
> |  | 2 | 0.904 | 0.899 | **0.907** |
> |  | 4 | 0.922 | **0.925** | 0.917 |
> |  | 8 | 0.928 | **0.933** | 0.924 |
> | Vehicle | 1 | 0.661 | 0.418 | **0.696** |
> |  | 2 | 0.722 | 0.523 | **0.753** |
> |  | 4 | 0.711 | 0.588 | **0.733** |
> |  | 8 | 0.729 | 0.688 | **0.766** |
>
> [1] Differentially Private Learning Needs Better Features (or Much More Data), ICLR 2021
>
> [2] DPNAS: Neural Architecture Search for Deep Learning with Differential Privacy, AAAI 2022
>
> **L1. Limitation.**
>
> **Computation of the gradient method.** For the computational time of gradient methods, we show the results in Table 4. For gradient methods, GRPUA performs c separate binary classifications and therefore takes several times longer than ours-GP, which updates all parameters simultaneously.
>
> **Additional limitations.** We will add the limitation in detail. For example, our framework is limited to linear feature spaces without kernel tricks, which fail to capture non-linear decision boundaries. Additionally, the scalability to large-scale datasets with thousands of classes and comprehensive evaluation against modern DP deep learning methods remains unexplored.
>
> **[General]** Additionally, we kindly request the reviewer to take a look at any interesting topics during the rebuttal period. In summary, we received positive feedback on: (i) reducing redundant privacy budget consumption for multi-class SVM in DP scenarios [Fj1B, orwF, hEqw, EHLs], (ii) empirical strengths [N89T, Fj1B, hEqw, EHLs], and (iii) well-written and well-executed paper [Fj1B, hEqw].
>
> For each rebuttal concerning “weakness” and “question”, we addressed the following topics:
>
> - [**N89T**] Our novelty and the importance of single data access in DP, and details of theorems
> - [**Fj1B**] Clarity of Lemma 1, convexity analysis of Eq. (12), and selection of hyperparameter $R$
> - [**orwF**] DPSVM in LLM era, selection of all-in-one base model, and advanced techniques in DP-SGD
> - [**hEqw**] Sub-sampling in gradient descent, low performance, and limitations
> - [**EHLs**] Inefficiency, additional comparison methods/related works, membership inference attack, and realistic impacts
>
> If the reviewer has any questions about these topics, please refer to our rebuttals for the corresponding reviewers.

---

> > ### Comment · Reviewer_hEqw · 2025-08-04
> > **Thank you**
> >
> > Thanks for the thorough rebuttal. I think you should include more batch sizes to shed light on the optimal value, which could be larger than 128. The rebuttal confirms my leaning towards acceptance.

---

### Official Review · Reviewer_orwF · 2025-06-25

**Clarity:** 4
**Significance:** 2
**Originality:** 3
**Rating:** 5
**Confidence:** 4

**Summary:**

This work focuses on using differential privacy for SVM. The methods in the binary case are not applicable. Also, the existing methods for multi-class classification require multiple binary SVMs with repeated data accesses for repeated consumption of the privacy budget. To address so, this work utilizes the all-in-one SVMs, which allows a single access to each data.

**Questions:**

1. You used CS-SVM for the all-in-one SVM in this work. Can you also explain more why you use it and compare it with WW-SVM and M3-SVM by a table? It is ok to include such information in your appendix.
2. For gradient perturbation, is it possible to use techniques in the DP-SGD approaches to enhance PMSVM?

**Ethical Concerns:**

["NO or VERY MINOR ethics concerns only"]

**Final Justification:**

Although SVM is outdated for the LLM era, this paper is a valuable contribution to the field of SVM with DP. Hence, it is probably worths for some real-world downstream tasks, which may not have the deployment ability of LLMs. To conclude, I agree to accept it.

**Quality:**

3

**Strengths And Weaknesses:**

Strength:

1. All-in-one SVMs utilized in this work avoid redundant data accesses compared to existing one-vs-rest and one-vs-one approaches for multi-class DP-SVM, which reduce the redundant consumption for privacy budget.
2. This work provides DP guarantees for weight perturbation by using leave-one-out bound and extends to multi-class case with low cost.
3. With moments accountant, this work lowers the privacy cost of the composition in gradient perturbation. Also Theorem 3 guarantees the post-processing property.

Weakness:

1. The SVM used as a multi-class classifier in the LLM era is relatively out-dated, hence the importance of DP-SVM is probably not significant, although this work advances the traditional ML with privacy-enhance techniques.

---

> ### Author Rebuttal · Authors · 2025-07-30
>
> First of all, we thank the reviewer for acknowledging the novelty of our paper. We now answer the question individually.
>
> **W1 Traditional ML in LLM Era**
>
> We fully agree with the reviewer’s concern about the importance of traditional ML in the LLM era. As researchers in ML and privacy, we believe the study of traditional ML techniques remains crucial, even if they may be undervalued today. Traditional methods such as SVM, regression, and clustering can still offer practical advantages. With small datasets, deep learning performance is not guaranteed. Moreover, privacy is especially important with small and limited data, since DP addresses the effect of changing a single sample.
>
> In this perspective, we can design a hybrid approach in the LLM era. As LLM‑based agents and multi‑modal models advance, designing automated pipelines for privacy‑preserving learning becomes feasible. For example, an LLM agent could select training strategies for private data: using DP deep learning with large models when data is sufficient, or choosing smaller ML models when data is limited or when training time or model complexity is constrained.
>
> We believe this line of research on lightweight machine learning models remains valuable in the LLM era. We authors are actively considering future direction and welcome any suggestions from the reviewer regarding this paper or potential follow‑up work.
>
> **Q1. Selection of base models in all-in-one SVMs**
> We agree with the reviewer’s question about using other all‑in‑one methods. As explained on line 102, different methods handle the maximum margin differently and, consequently, use different slack variables.
>
> We clarify that we use CS‑SVM for weight perturbation (subsection 3.2) and M3‑SVM for gradient perturbation (subsection 3.3) in our DP all‑in‑one SVMs. Because Equation 8 provides a unified view of all‑in‑one SVMs, Theorem 1 can also be applied to other variants, as the reviewer suggested. However, computing the largest eigenvalue of the Gram matrix $\lambda_{max}$ is difficult for those models; CS‑SVM allows easier calculation due to its orthogonal‑basis properties. Note that CS-SVM is the only method officially supported by the sklearn SVM package, thus having the ease of implementation for paper readers.
>
> For gradient perturbation, all other all‑in‑one SVMs can likewise be formulated with Huber‑style losses, similar to Equation 12. We choose M3‑SVM because it performs well with gradient descent, as shown in [1].
>
> We recognize that readers may wish to understand our choice of base models, and we will add a more detailed explanation in the Appendix.
>
> [1] Multi-Class Support Vector Machine with Maximizing Minimum Margin, AAAI 2024.
>
> **Q2. Advanced techniques in DP-SGD**
>
> It is true that since the development of DP-SGD in 2016, clipping‑based gradient updates in DP‑SGD is that these approaches are extensively studied, allowing us to leverage recent analytical techniques for gradient‑based optimization. As noted on line 228, we use Opacus, a modern library developed by Meta Open Source. Opacus is actively updated with new DP‑SGD algorithms, including optimizers and sampling strategies.
>
> For the adaptive gradient perturbation in Equation 16, we employ the DP‑Adam [1] optimizer, which is officially supported by the Opacus library. Thus, adopting advanced techniques is also feasible within our framework. When we consider the private gradient of Theorem 2 as $\hat {\mathbf g}_ t$, the adaptive update formulation is as follows:
>
> $$\hat{\mathbf m} _ t = \beta_1 \hat{\mathbf m} _ {t-1} + (1-\beta_1)\hat{\mathbf g} _ t,\ \hat{\mathbf v} _ t = \beta_2 \hat{\mathbf v} _ {t-1} + (1-\beta_2)(\hat{\mathbf g} _ t \odot \hat{\mathbf g} _ t)$$
>
> $$\hat{\mathbf m} _ t = \frac{\hat{\mathbf m} _ t}{1-\beta_1^t},\ \hat{\mathbf v} _ t = \frac{\hat{\mathbf v} _ t}{1-\beta_2^t},\ \hat  {\mathbf w} _ t = \hat {\mathbf w} _ {t-1} - \eta\frac{\hat{\mathbf m} _ t}{\sqrt{\hat{\mathbf v} _ t} + \gamma}.$$
>
> Furthermore, for sub-sampling methods, we first clarify that we used a batch size of 128 rather than full batch gradient descent. We apologize for this confusion and will clarify the use of subsampling methods in the paper.
> We utilized a batch size of 128 rather than using a full batch for gradient descent. By utilizing Opacus, we also include recent Poisson subsampling for mini-batch update. The paper [2] suggested by the reviewer [hEqw] provides mathematical guarantees for the composition method (which performs better than basic composition in Remark 1 or moment accounting in Lemma 2) for subsampling approaches. Compared to using full batch gradient descent, a moderate batch size, such as 64 or 128, shows better performance ($\\approx$ means the mean batch size of Poisson sampling).
>
> | Dataset | ε | bs$\\approx$32 | bs$\\approx$64 | bs$\\approx$128 | bs=full |
> | --- | --- | --- | --- | --- | --- |
> | Cornell | 1 | 0.478±0.112 | 0.681±0.000 | **0.693±0.032** | 0.480±0.066 |
> |  | 2 | 0.564±0.021 | 0.687±0.000 | **0.707±0.023** | 0.590±0.024 |
> |  | 4 | 0.641±0.058 | 0.737±0.015 | **0.752±0.023** | 0.645±0.034 |
> |  | 8 | 0.667±0.019 | 0.761±0.004 | **0.765±0.024** | 0.671±0.007 |
> | Dermatology | 1 | 0.297±0.059 | 0.883±0.077 | **0.905±0.017** | 0.243±0.143 |
> |  | 2 | 0.369±0.068 | 0.937±0.034 | **0.951±0.042** | 0.320±0.021 |
> |  | 4 | 0.599±0.056 | 0.919±0.023 | **0.978±0.012** | 0.541±0.166 |
> |  | 8 | 0.712±0.067 | 0.964±0.028 | **0.976±0.018** | 0.635±0.059 |
> | HHAR | 1 | 0.878±0.009 | **0.934±0.003** | 0.929±0.007 | 0.884±0.005 |
> |  | 2 | 0.908±0.002 | 0.941±0.004 | **0.946±0.004** | 0.913±0.008 |
> |  | 4 | 0.928±0.001 | 0.954±0.004 | **0.956±0.006** | 0.932±0.005 |
> |  | 8 | 0.940±0.000 | 0.950±0.002 | **0.959±0.003** | 0.943±0.003 |
> | ISOLET | 1 | 0.059±0.010 | 0.465±0.042 | **0.501±0.025** | 0.037±0.014 |
> |  | 2 | 0.054±0.022 | 0.614±0.013 | **0.687±0.017** | 0.037±0.005 |
> |  | 4 | 0.124±0.034 | 0.769±0.039 | **0.804±0.010** | 0.105±0.030 |
> |  | 8 | 0.120±0.013 | 0.834±0.018 | **0.840±0.013** | 0.144±0.045 |
> | USPS | 1 | 0.813±0.002 | **0.918±0.002** | 0.897±0.006 | 0.829±0.008 |
> |  | 2 | 0.855±0.011 | **0.922±0.001** | 0.907±0.006 | 0.854±0.009 |
> |  | 4 | 0.876±0.004 | **0.920±0.003** | 0.917±0.002 | 0.874±0.005 |
> |  | 8 | 0.891±0.002 | **0.931±0.003** | 0.924±0.003 | 0.887±0.010 |
> | Vehicle | 1 | 0.363±0.163 | 0.688±0.024 | **0.696±0.060** | 0.424±0.064 |
> |  | 2 | 0.410±0.094 | 0.684±0.024 | **0.753±0.007** | 0.480±0.075 |
> |  | 4 | 0.524±0.080 | 0.718±0.018 | **0.733±0.023** | 0.571±0.031 |
> |  | 8 | 0.651±0.056 | 0.731±0.007 | **0.766±0.009** | 0.639±0.040 |
>
> We will include this result in the paper and clarify the use of the subsampling method.
>
> [1] Large Language Models Can Be Strong Differentially Private Learners, ICLR 2022
>
> [2] Composition of Differential Privacy & Privacy Amplification by Subsampling, Arxiv (2022)
>
>
> **[General]** Additionally, we kindly request the reviewer to take a look at any interesting topics during the rebuttal period. In summary, we received positive feedback on: (i) reducing redundant privacy budget consumption for multi-class SVM in DP scenarios [Fj1B, orwF, hEqw, EHLs], (ii) empirical strengths [N89T, Fj1B, hEqw, EHLs], and (iii) a well-written and well-executed paper [Fj1B, hEqw].
>
> For each rebuttal concerning “weakness” and “question”, we addressed the following topics:
>
> - [**N89T**] Our novelty and the importance of single data access in DP, and details of theorems
> - [**Fj1B**] Clarity of Lemma 1, convexity analysis of Eq. (12), and selection of hyperparameter $R$
> - [**orwF**] DPSVM in LLM era, selection of all-in-one base model, and advanced techniques in DP-SGD
> - [**hEqw**] Sub-sampling in gradient descent, low performance, and limitations
> - [**EHLs**] Inefficiency, additional comparison methods/related works, membership inference attack, and realistic impacts
>
> If the reviewer has any questions about these topics, please refer to our rebuttals for the corresponding reviewers.

---

> > ### Comment · Reviewer_orwF · 2025-08-06
> >
> > Thanks for your response! It clears all my questions.

---

### Official Review · Reviewer_Fj1B · 2025-07-01

**Clarity:** 3
**Significance:** 3
**Originality:** 3
**Rating:** 5
**Confidence:** 3

**Summary:**

This paper studied how DP can be properly applied to multi-class SVMs. The authors presents a novel privacy-preserving multi-class SVM framework designed to mitigate the issue of repeated data access found in existing multi-class SVM approaches under DP scenarios. By employing all-in-one methods, their framework significantly reduces the noise level through decreased data access, eliminating the need for multiple binary classifiers for both weights and gradient. They provide rigorous sensitivity and convergence analyses to ensure DP in all-in-one SVMs. Empirical results demonstrate that our approach surpasses existing DP-SVM methods in multi-class scenarios.

**Questions:**

Please refer to the weakness section for questions.

**Ethical Concerns:**

["NO or VERY MINOR ethics concerns only"]

**Final Justification:**

The authors have addressed my concerns that include:

(1) Give a clear introduction on how they use Lemma 1 and Theorem 1 to guarantee the validity of the proposed method.

(2) Show a detailed proof of the convexity of Eq. (2), which ensures the convergence of the loss function to the global minimum

(3) Provide a concise and insightful analysis on the gap of computational time between OPERA and PMSVM-WP

**Limitations:**

Yes

**Quality:**

3

**Strengths And Weaknesses:**

__Strength__

This paper studied multi-class classification using DP-SVM, which is not actively investigated.

1. To address the issue of multiple data accesses in multi-class classification, they introduce the differentially Private Multi-class SVM (PMSVM) based on the all-in-one multi-class SVM framework. This approach allows to access each data point only once.

2. Their methods include two differentially private variants: PMSVM with weight perturbation and PMSVM with gradient perturbation.

3. In summary, the proposed PMSVM framework reduces the number of data accesses per sample and achieves better utility-privacy trade-off.

4. Empirical results show the practicality of the proposed methods.

The paper is logically written and well-organized. The empirical result (_e.g._, Figure) supports their claims.

__Weakness__

Here are some questions that I think may help improve this paper.

1. Line 92: the preprositional phrase _Instead of calculating support vectors for each support vector, ..._ is confusing. Could the authors clarify what is meant here?

2. Line 94: The $y_{i,kl}$ appears only once. What does this term mean?

3. The first two constraints in Eq. (7) are ambiguous. _e.g._ What does $\xi_{ikl}(y_i = k)$ represent?

4. Lemma 1 presents a leave-one-out bound for SVM. Could the authors clarify the form of the input scaler $g(\cdot)$ and the convex function $T$ look like in the binary support vector machine defined in Eq. (4)?

5. Line 143: The $\sigma_w$ does not appear in Eq. (3). Instead, it first appears in Remark 3.

6. What is the difference between $\hat{w}_t$ and $w_t$ in Definition 4 or Lemma 3? Is it a typo?

7. Line 164: Could the authors give a brief proof on the convexity of Eq. (12)?

8. Line 175: The hyperparameter $R$ is fixed at 1. Is there any guiding principle for selecting an appropriate value of $R$?

9. It is well-known that strict convexity does not necessarily imply strong convexity. Could the author provide a brief proof to demostrate the strong convexity of Eq. (12)?

10. Lines 221 and 224: Is the $\varepsilon$ in $\varepsilon=4$ equivalent to the $\epsilon$ defined as the privacy budget?

11. Why is the difference significant between Ours-WP and OPERA on ISOLET? Could the authors give some insights?

---

> ### Author Rebuttal · Authors · 2025-07-30
>
> First of all, we thank the reviewer for reading our paper carefully and suggest various ways to improve the quality of our paper. We now answer the question individually.
>
> **W1.  Line 92 “*Instead of calculating support vectors for each support vector”***
>
> Thank you for advising on the clarity of the proposed paper. We agree that this phrase might confuse the readers. We will change it to “***Instead of calculating support vectors for each binary classification”.***
>
> **W2. Line 94 notation $y_{i,kl}$**
>
> It means the label of $y_{i}$, when we consider the binary classification of class $k$ and $l$, similar to line 94 “$\upsilon_{y_i,k}$ equals 1 if $k=y_i$ and 0 otherwise.”
>
> However, the notation is used only once, and the $y_i, i \in [n]$ indicates a multi-class label in Equation 5; Thus, we remove the notation of $y_{i,kl}$.
>
> **W3. Equation 7**
>
> As the reviewer suggested, we will revise Equation (7) as follows:
>
> \begin{cases}
> f_{kl}(\mathbf x_i) \ge 1 - \xi_{ikl}, & \text{where } y_i = k \\\\
> f_{kl}(\mathbf x_i) \le -1 + \xi_{ikl}, & \text{where } y_i = l
> \end{cases}
>
> **W4. Lemma 1 compared to the binary case**
>
> We apologize for the confusion. Firstly, we explain how the current Lemma deals with the binary SVM case and decide to change the Lemma with the proof part of the all-in-one case proof of Theorem 1, which is now in the Appendix.
>
> **The clarity of the current Lemma 1 in binary SVM.** Lemma 1 is a generalized version of the leave-one-out bound for kernel methods [1] and can be applied to cover binary SVM, since the proof of the lemma is for general case using subgradient of the convexity of $T$ and the definition of optimal $\tilde{\alpha}$ and $\tilde{\alpha}^{[k]}$.
> Specifically, let transform Eq. (4)
>
> $\max_{\alpha} \sum_{i=1}^{n} \alpha_i - \frac{1}{2} \sum_{i=1}^{n} \sum_{j=1}^{n} \alpha_i \alpha_j y_i y_j x_i^\top x_j =  \min_{\alpha}  \frac{1}{2} \sum_{i=1}^{n} \sum_{j=1}^{n} \alpha_i \alpha_j y_i y_j x_i^\top x_j - \sum_{i=1}^{n} \alpha_i$.
>
> (We skip bold for vector due to the limit.) In this setting, $g(x) = x$ and $T(-\alpha, x)$ is convex (affine) with respect to $\alpha$.
>
> With this formula, the lemma can be modified as:
>
> $\min_{\alpha}  \frac{1}{2} \sum_{i=1}^{n} \sum_{j=1}^{n} \alpha_i \alpha_j y_i y_j g(x_i)^\top g(x_j) +\sum_{i=1}^{n} T(-\alpha_i, x_i)$
>
> This version of Lemma 1 can be proved in the same way as the original proof:
>
> ---
> proof:
>
> Since $-\nabla_1 T(-\tilde{\alpha}_i, x_i) +  \sum _{j=1}^{n}\tilde{\alpha}_j y_i y_j  g(x_i)^\top g(x_j) = 0$,
>
> $-\nabla_1 T(-\tilde{\alpha}_i, x_i)(\tilde{\alpha}_i^{[k]}-\tilde{\alpha}_i) +  \sum _{j=1}^{n} \tilde{\alpha}_j y_i y_j g(x_i)^\top g(x_j)(\tilde{\alpha}_i^{[k]}-\tilde{\alpha}_i) = 0$. By the definition of subgradient, we have
>
> $-\nabla_1 T(-\tilde{\alpha}_i, x_i)(\tilde{\alpha}_i^{[k]}-\tilde{\alpha}_i) \leq T(-\tilde{\alpha}_i^{[k]},x_i)-T(-\tilde{\alpha}_i, x_i)$.
>
> Therefore, $T(-\tilde{\alpha}_i, x_i)- \sum _{j=1}^{n} \tilde{\alpha}_j y_i y_j g(x_i)^\top g(x_j)(\tilde{\alpha}_i^{[k]}-\tilde{\alpha}_i) \leq T(-\tilde{\alpha}_i^{[k]}, x_i)$.
>
> Summing over $i$, utilizing the definition of $\tilde{\alpha}_i^{[k]}$, and denote $\tilde{\alpha}_i^{[n]}=0$ as in the original proof. WLOG $k = n$, and since $y_i^2 =1$, then we can obtain:
>
> $\sum_{i=1}^{n}\sum_{j=1}^{n} y_iy_jg(x_i)^\top g(x_j)(\tilde{\alpha}_i^{[k]}-\tilde{\alpha}_i) (\tilde{\alpha}_j^{[k]}-\tilde{\alpha}_j) \leq \tilde{\alpha}_n^2 ||g(x_n)||_2^2$
>
> Therefore, for $\tilde{w}_D =\sum _{i=1}^{n} \tilde{\alpha}_i y_i g(x_i)$, we can get  $||{\tilde{w}_D}-{\tilde{w} _{D^k}}||_2 \leq |\tilde{\alpha} _{k}| ||g(x_k)||_2.$
>
> **New Lemma 1.** As pointed out by reviewer [N89T], we do not utilize Lemma 1 directly, *but just adopt the proof strategy.* To avoid confusion, we will remove Lemma 1 (originally from [1]) and replace it with a new lemma that we have proved in the proof of Theorem 1. The lemma specific to our all-in-one SVM can be stated as follows:
>
> ---
>
> For a convex function $T$, a dataset $D$, and input scaler $g(\cdot)$, let $\tilde{w}_D=\sum _{i=1}^n \tilde{\alpha}_i g(x_i)$ where $(\tilde{\alpha}_1,\ldots,\tilde{\alpha}_n)$ *is the solution to:* $\min _{{\alpha}} \left(\frac{1}{2} \sum _{i,p}\sum _{j,q}\sum _{k}\alpha _{i,p}\alpha _{j,q}\nu _{y_i,p}\nu _{y_j,q} g(x_i)^\top g(x_j)+\sum _{i,p} T \left(-\alpha _{i,p}\right)\right).$
> Let $D^n$ be $D$ with the $n$-th point $x_n$ removed, and let $\tilde{w} _{D^n}$ be defined similarly. Then the difference of the weights between original and leave-one-out SVMs is bounded as:
> $\sum _{k=1}^c ||\tilde{w} _k^{[n]} - \tilde{w} _k||^2 \leq \lambda _{\max}(G) |\tilde{\alpha} _{n}\|^2\|g(x_n)||^2.$
>
> ---
>
> We believe that this change makes the connection between Lemma  and Theorem clear.
>
> [1] Leave-one-out bounds for kernel methods. *Neural computation (2003).*
>
> **W5, W6, W10. Typos**
>
> Thank you. It is obvious typos. We will revise it.
>
> For W6, we will also include the following rules for readability concerning the notations of $\mathbf{\cdot}, \tilde{\cdot}, \hat{\cdot}$.
>
> $\mathbf{\cdot}$: general case such as subsection 2.2
>
> $\tilde{\cdot}$: optimal (or empirical) weight without noise addition (as in Definition 3)
>
> $\hat{\cdot}$: perturbed (thus private) weight with noise addition
>
>
>
> **W7, W9. Convexity of Eq. (12)**
>
> **Convexity**. We first prove the convexity of Eq. (12). We follow the proof of [1], detailed in Appendix A.1.
>
> $$
> \min_{W,\mathbf{b}}  \sum _{i=1}^{n}\sum _{k\neq y_i}        \frac{\gamma _{ik}+\sqrt{\gamma _{ik}^{2}+\varsigma^2}}{2}  +    \frac{C}{n}\sum _{k<l}||\mathbf{w}_k-\mathbf{w}_l|| _{2}^{2}
> $$
>
> For the first term, it is obvious that the hinge loss term is convex. In detail,
>
> $$
> g(x)=\frac{x+\sqrt{x^2+\delta^2}}{2},
> \quad
> g''(x)=\frac{\delta^2}{4(x^2+\delta^2)^{3/2}}\ge0,
> $$
>
> $$
> \gamma_{ik}(W,b)
> =1-\bigl(w_{y_i}^\top x_i - w_k^\top x_i + b_{y_i}-b_k\bigr),
> \quad
> \gamma_{ik}\bigl(\mu(W,b)+(1-\mu)(V,c)\bigr)
> =\mu\gamma_{ik}(W,b)+(1-\mu)\gamma_{ik}(V,c),
> $$
>
> $$
> f_1(W,b)
> =\sum_{i=1}^n\sum_{k\neq y_i}g\bigl(\gamma_{ik}(W,b)\bigr),
> \quad
> f_1\bigl(\mu(W,b)+(1-\mu)(V,c)\bigr)
> \le\mu f_1(W,b)+(1-\mu) f_1(V,c).
> $$
>
> For the second term, it is also convex because
>
> $$
> \mathcal{M}(\mu W + (1-\mu)V)
> = \sum _{k=1}^{c-1}\sum _{l=k+1}^{c}
> h\bigl(\mu(\mathbf{w}_k - \mathbf{w}_l) + (1-\mu)(\mathbf{v}_k - \mathbf{v}_l)\bigr)
> \le \sum _{k=1}^{c-1}\sum _{l=k+1}^{c}
> \bigl(\mu h(\mathbf{w}_k - \mathbf{w}_l) + (1-\mu) h(\mathbf{v}_k - \mathbf{v}_l)\bigr)
> = \mu\mathcal{M}(W) + (1-\mu)M(V).
> $$
>
> Detailed analysis is in Appendix A.1 of [1].
>
> **Strong convexity.**
> It is true that the (strict) convexity does not guarantee strong convexity; thus, the theoretical analysis of Theorem 3 might not be true. The Hessian of both the first and second terms of Equation 12 is positive semi-definite, and thus can be zero. For strong convexity, we need a **positive definite** Hessian matrix.
>
> Specifically, for the first term, since $g''(x) = \frac{\delta^2}{4(x^2+\delta^2)^{3/2}} \to 0$ as $|x| \to \infty$ and $\gamma_{ik}$ is linear in $W$, the Hessian can vanish along certain directions. For the second term, the pairwise regularization $\sum_{k<l}\|\mathbf{w}_k-\mathbf{w}_l\|_2^2$ has a null space corresponding to uniform translations $\mathbf{w}_k \to \mathbf{w}_k + \mathbf{v}$ for all $k$, yielding $\mathbf{v}^T\nabla^2 f_2\mathbf{v} = 0$ for $\mathbf{v} = [\mathbf{u}, \mathbf{u}, \ldots, \mathbf{u}]^T$. Thus, both terms have a **positive semi-definite** Hessian.
>
> We thus revisit [1] and revised Equation 12 to have a regularization term $\mu(||{W}||^2_F+||\mathbf{b}||_2^2)$ term (Eq 25 in [1]). As the regularizer is the sum of quadratic terms w.r.t. $W$ and $\mathbf{b}$, it ensures a **strong convexity**.
>
> $$
> \min_{W,\mathbf{b}}  \sum _{i=1}^{n}\sum _{k\neq y_i}        \frac{\gamma _{ik}+\sqrt{\gamma _{ik}^{2}+\varsigma^2}}{2}  +    \frac{C}{n}\sum _{k<l}||\mathbf{w}_k-\mathbf{w}_l|| _{2}^{2} + \mu(||{W}||^2_F+||\mathbf{b}||_2^2)
> $$
>
> Then, we can ensure the total Hessian is $\mathbf{x}H\mathbf{x}>0$ for $\forall \mathbf{x} \neq 0$ with a positive definite regularization term, thus strong convexity. We will revise the gradient updates in Theorem 2 correspondingly.
>
> **Experiments.** In experiments, as we used the base code of [1], we already used $\mu>0$ (para.lam in main-opacus.py in supplementary code, e.g., 0.0005 for HHAR, 0.005 for Cornell, …).
>
> We really appreciate the reviewer of this analysis.
>
> [1] Multi-class Support Vector Machine with Maximizing Minimum Margin, AAAI 2024.
>
> **W11. Ours-WP and OPERA on ISOLET**
>
> We understand this question concerns the runtimes in Table 4. If not, please let us know.
>
> Regarding runtime between OPERA and PMSVM‑WP, we use scikit‑learn’s “LinearSVC,” which implements the OvR method for multi‑class classification for OPERA. In contrast, PMSVM‑WP uses the “crammer_singer” option.
> Thus, the difference relies only on the implementation of two methods in sklearn because the proposed weight perturbation requires only an additional single noise‑addition step: $O(1)$ per parameter, i.e., $O(\text{num of params})$ overall.
>
> To explain the computational part, we can think that the Crammer–Singer formulation solves a single joint QP with $nc$ variables, whereas OvR decomposes into $c$ independent binary SVMs, each with $n$ variables. Given that standard QP solvers scale as $O(\text{num of params}^3)$, Crammer–Singer entails $O(n^3c^3)$, while OvR requires $O(cn^3)$. In practice (e.g., in scikit‑learn using LIBLINEAR), the observed gap is about $O(c)$ times practically. This explains the runtime gap between OPERA and PMSVM‑WP. We will add this discussion to the main paper and the appendix on weight‑perturbation computation.
>
> On the other hand, for gradient perturbation, because the model is updated in a single pass rather than via sequential OvR training, PMSVM‑GP is superior to GRPUA in both performance and computational efficiency.
>
> **We will continue the response of W8. hyperparameter $R$ in the comments below.**

---

> > ### Author Response · Authors · 2025-08-01
> >
> > **W8. hyperparameter $R$**
> >
> > The hyperparameter $R$ denotes the clipping threshold for gradient clipping. In DP‑SGD for deep learning models, there is no universal rule for selecting the clipping value $R$ (often denoted by $C$ in DP‑SGD). However, as $R$ increases, the level of Gaussian noise also increases; using a larger $R$ is known to harm performance in DP‑SGD [1]. See Section B.1. We follow their basic setup with $R=1$.
> >
> > To experimentally test the difference between selecting $R$, we tested on R=0.01, 0.1, 1, and 10. It is true that there is no universal rule to choose $R$, we concluded that $R=1$ from [1] is quite reasonable for the gradient method in our paper.
> >
> > | Dataset | ε | R=0.01 | R=0.1 | R=1 | R=10 |
> > | --- | --- | --- | --- | --- | --- |
> > | Cornell | 1 | 0.677±0.007 | 0.681±0.000 | **0.693±0.032** | 0.347±0.110 |
> > |  | 2 | 0.679±0.003 | 0.687±0.006 | **0.707±0.023** | 0.560±0.034 |
> > |  | 4 | 0.683±0.003 | 0.745±0.017 | **0.752±0.023** | 0.653±0.025 |
> > |  | 8 | 0.747±0.006 | **0.765±0.024** | **0.765±0.024** | 0.657±0.006 |
> > | Dermatology | 1 | 0.842±0.028 | 0.878±0.070 | **0.905±0.017** | 0.171±0.110 |
> > |  | 2 | 0.950±0.016 | **0.955±0.028** | 0.951±0.042 | 0.230±0.084 |
> > |  | 4 | **0.987±0.000** | 0.978±0.016 | 0.978±0.012 | 0.559±0.034 |
> > |  | 8 | **0.982±0.008** | 0.978±0.016 | 0.976±0.018 | 0.743±0.036 |
> > | HHAR | 1 | 0.922±0.004 | **0.929±0.002** | **0.929±0.007** | 0.885±0.007 |
> > |  | 2 | 0.943±0.001 | **0.947±0.002** | 0.946±0.004 | 0.913±0.004 |
> > |  | 4 | 0.948±0.001 | 0.951±0.002 | **0.956±0.006** | 0.931±0.002 |
> > |  | 8 | 0.953±0.001 | 0.953±0.001 | **0.959±0.003** | 0.938±0.003 |
> > | ISOLET | 1 | 0.431±0.007 | 0.458±0.013 | **0.501±0.025** | 0.057±0.030 |
> > |  | 2 | 0.661±0.027 | 0.662±0.026 | **0.687±0.017** | 0.076±0.029 |
> > |  | 4 | 0.732±0.013 | 0.746±0.010 | **0.804±0.010** | 0.119±0.022 |
> > |  | 8 | 0.825±0.024 | **0.849±0.014** | 0.840±0.013 | 0.110±0.002 |
> > | USPS | 1 | 0.917±0.003 | **0.920±0.001** | 0.897±0.006 | 0.810±0.004 |
> > |  | 2 | 0.922±0.001 | **0.927±0.000** | 0.907±0.006 | 0.856±0.003 |
> > |  | 4 | 0.927±0.002 | **0.929±0.003** | 0.917±0.002 | 0.873±0.002 |
> > |  | 8 | 0.928±0.002 | **0.931±0.002** | 0.924±0.003 | 0.891±0.003 |
> > | Vehicle | 1 | 0.641±0.026 | 0.680±0.050 | **0.696±0.060** | 0.329±0.010 |
> > |  | 2 | 0.684±0.017 | 0.716±0.009 | **0.753±0.007** | 0.484±0.019 |
> > |  | 4 | 0.710±0.015 | 0.727±0.014 | **0.733±0.023** | 0.578±0.071 |
> > |  | 8 | 0.722±0.009 | 0.741±0.020 | **0.766±0.009** | 0.673±0.038 |
> >
> > [1] Unlocking High-Accuracy Diﬀerentially Private Image Classification through Scale, 2022.
> >
> > **[General]** Additionally, we kindly request the reviewer to take a look at any interesting topics during the rebuttal period. In summary, we received positive feedback on: (i) reducing redundant privacy budget consumption for multi-class SVM in DP scenarios [Fj1B, orwF, hEqw, EHLs], (ii) empirical strengths [N89T, Fj1B, hEqw, EHLs], and (iii) well-written and well-executed paper [Fj1B, hEqw].
> >
> > For each rebuttal concerning “weakness” and “question”, we addressed the following topics:
> >
> > - [**N89T**] Our novelty and the importance of single data access in DP, and details of theorems
> > - [**Fj1B**] Clarity of Lemma 1, convexity analysis of Eq. (12), and selection of hyperparameter $R$
> > - [**orwF**] DPSVM in LLM era, selection of all-in-one base model, and advanced techniques in DP-SGD
> > - [**hEqw**] Sub-sampling in gradient descent, low performance, and limitations
> > - [**EHLs**] Inefficiency, additional comparison methods/related works, membership inference attack, and realistic impacts
> >
> > If the reviewer has any questions about these topics, please refer to our rebuttals for the corresponding reviewers.

---

> > ### Comment · Reviewer_Fj1B · 2025-08-04
> >
> > Thanks very much for the authors' response. I'll increase the score accordingly.

---

> > > ### Author Response · Authors · 2025-08-05
> > > **Thank you for comment**
> > >
> > > We are grateful for the reviewer’s careful and comprehensive comments on our rebuttal.
> > >
> > > Again, we sincerely appreciate the reviewer's detailed analysis and feedback on our paper to modify confusing parts and provide theoretical clarity.
> > >
> > > We believe that our work has significantly improved.

---

### Official Review · Reviewer_N89T · 2025-07-02

**Clarity:** 3
**Significance:** 3
**Originality:** 2
**Rating:** 4
**Confidence:** 3

**Summary:**

This paper studies the problem of ensuring differential privacy for training data in multi-class SVMs. Specifically, it proposes private training algorithms for multi-class SVMs: one based on weight perturbation (by solving the empirical risk minimization problem), and another based on gradient perturbation. The paper also analyzes the privacy guarantees of both methods.

**Questions:**

**1. Clarification on utility advantage in Theorem 3:** How does Theorem 3 demonstrate a utility advantage over existing approaches?

**2. Technical contribution of Theorem 1:** Theorem 1 appears to be a direct application of Lemma 1 and the DP-ERM framework to the standard multi-class SVM setting. What is the technical novelty of this result? The connection between these components seems straightforward and, to a large extent, already known.

**3. Novelty of Theorem 2:** The gradient perturbation method described in Theorem 2 appears to be a specific instance of the well-established DP-SGD framework, which is broadly applicable to many learning problems. What, then, is the novel contribution of Theorem 2 in this context?

**4. Notation in Equation (7):** The use of notation such as $(y_i = k)$ and $(y_i = l)$ in Equation (7) is unclear and should be clarified.

**5. Access frequency vs. privacy-utility trade-off:** What is the precise relationship between the number of accesses per sample and the privacy-utility trade-off? This connection is not clearly articulated in the current manuscript.

6. What does $\approx$ in (16) mean precisely?

**Typo:** There appears to be a typo in the definition of the function $T(\cdot, \cdot)$ on Line 377.

While I am open to revising my evaluation if the authors can provide convincing responses to these points, I currently believe that the theoretical contributions of the paper are insufficient for acceptance.

**Ethical Concerns:**

["NO or VERY MINOR ethics concerns only"]

**Final Justification:**

I thank the authors for taking note of my concerns, and for responding to them in a detailed manner.
Authors have addressed my concerns and have suggested revisions that would improve the quality (i.e., positioning with respect to prior work, technical writing and discourse) of the paper.

Taking into account the authors' rebuttal, and other reviewers' comments, I am happy to increase my score to 4.

**Limitations:**

Yes.

**Paper Formatting Concerns:**

None.

**Quality:**

3

**Strengths And Weaknesses:**

**Strengths:**

1. The paper designs differential privacy (DP) mechanisms tailored for general all-in-one multi-class SVMs.
2. It includes comprehensive empirical evaluations.

**Weaknesses:**

**1. Limited novelty.** The contributions appear to be straightforward applications of existing general results to the specific setting of multi-class SVMs, largely drawn from prior work.

**2. Lack of formal discussion on improved privacy-utility trade-off.** Although the abstract and introduction claim improvement in the privacy-utility trade-off, the paper does not seem to present a formal or substantial treatment of this aspect.

**3. Heavily based on prior work.** Much of the content is a recollection or adaptation of existing results, with minimal original contributions.

---

> ### Author Rebuttal · Authors · 2025-07-30
>
> Thank you for your insightful questions. Below, we first address the major concerns [W1, W3, Q5] and then respond to the remaining points. We kindly ask that you reconsider your score after reviewing our detailed responses.
>
> **W1 Limited novelty.**
>
> We acknowledge the reviewer's concern about novelty. However, as highlighted by other reviewers [N89T, orwF, EHLs], we strongly believe our paper presents significant novelty through the lens of differential privacy. As detailed in our Introduction and Related Works sections, following the initial DP-SVM proposal [1] or general ERM case [2], subsequent research focused exclusively on binary DP-SVM cases, leaving the multi-class scenario unexplored. However, as discussed in subsection 3.1 and Table 1, **OvR or OvO methods necessarily increase per-sample data access for each class, severely impacting DP training effectiveness** (detailed in **Q5**).
>
> While [3] also identified this privacy-unfriendly issue with SVM for multi-class classification, they detoured this issue with a private support vector-based clustering without actually solving the problem. On the other hand, we provide a direct solution, which is orthogonal to the existing advantages of all-in-one SVM in a non-private scenario.
>
> [1] Learning in a large function space: Privacy-preserving mechanisms for SVM learning, Journal of Privacy and Confidentiality (2012)
>
> [2] Differentially private empirical risk minimization. Journal of machine learning research (2011)
>
> [3] Efficient differentially private kernel support vector classifier for multi-class classification. Information Sciences (2023)
>
> **W3 Heavily based on prior work.**
>
> Building on our **W1** response, we clarify that our method is not a simple combination of existing methods.
> First, implementing an all-in-one SVM in DP contexts is far from trivial. In a non-private scenario, it is hard to say that all-in-one methods are significantly preferred over OvR classifiers in practice. However,  in the privacy domain, we can dramatically reduce the privacy budget of multi-class classification to the same level as binary classification. This advantage appears in neither the previous DP-SVM nor the all-in-one SVM literature.
>
> Second, we introduce weight and gradient-based methods to guarantee DP for the all-in-one SVM.  Since SVM solves a fixed margin maximization problem using the specific objective function, previous works developed how to inject the noise within existing SVM formulations or develop a DP version of kernel tricks, not developing a new formulation of SVMs. To the best of our knowledge, existing DP-SVM research follows: within binary SVM formulations, (i) establishing efficient DP guarantees via noise injection in various components, and (ii) proving DP sensitivity, while minimally addressing multi-class scenarios.
>
> Thus, our paper follows the similar flow as follows: (i) demonstrating how binary SVM-based OvO and OvR negatively impact privacy budgets in multi-class settings, (ii) developing new methods for DP-SVM with all-in-one frameworks, and (iii) theoretically proving sensitivity and showing empirical advantages across diverse experiments.
>
> **Q5. Access frequency vs. privacy-utility trade-off**
>
> We appreciate the reviewer's question about the precise relationship between per-sample access frequency and the privacy-utility trade-off. This represents the core contribution of our paper, and we believe a thorough explanation will benefit both the reviewer and future readers. We will add the detailed explanation in the paper after the rebuttal.
>
> Our goal is to reduce the number of data accesses to individual data samples; thus, “All-in-one SVMs utilized in this work avoid redundant data accesses compared to existing one-vs-rest and one-vs-one approaches for multi-class DP-SVM, which reduce the redundant consumption for privacy budget,” noted by reviewer [orwF].
>
> The composition theorem, one of differential privacy's most fundamental properties, is presented in Remark 1 (line 65) for two mechanisms. When we consider the general composition for $(\epsilon,\delta)$‑d.p. algorithms for multiple $k\ge2$ mechanisms [1], we can similarly formulate the composition as follows (shorted version):
>
> ---
>
> Let $\mathcal{M_1}\colon D\to\mathcal{C_1}$ be $(\epsilon,\delta)$‑d.p., and for each $k\ge2$ let
> $\mathcal{M_k}\colon (D, s_{k-1},\dots,s_{1}) \to\mathcal{C_k}$
> be $(\epsilon,\delta)$‑d.p. for all given $(s_{k-1},\dots,s_{1})\in\bigotimes_{j=1}^{k-1}\mathcal{C_j}$.
> Then for any neighboring datasets $D,D'$ and any $S\subseteq\bigotimes_{j=1}^{k-1}\mathcal{C_j}$,
> $$\Pr\bigl((\mathcal{M_1},\dots,\mathcal{M_k})\in S\bigr)\le e^{k\epsilon}\Pr'\bigl((\mathcal{M_1},\dots,\mathcal{M_k})\in S\bigr)+k\delta$$
>
> ---
>
> Simply, the composition theorem indicates that the number of accesses to data $D$ directly affects the noise level if each mechanism has a dependency on training data. When applying this principle to the OvR case, the composition of c classifiers requires c$\epsilon$ privacy budget when each binary classifier requires $\epsilon$. Therefore, to maintain the same total privacy budget (e.g., $\epsilon$=8), we can only allocate $\frac{8}{c}$ to each binary classifier, amplifying the amount of noise on each classifier.
> In contrast, our method PMSVM requires only one data access to build a multi-class classifier, allowing us to utilize the full privacy budget $\epsilon$. This is why we propose a multi-class DP SVM based on the all-in-one method.
>
> Thanks to the reviewer’s comment, we will change Remark 1 into the above formulation and add a detailed explanation about the precise relationship between per-sample access frequency and the privacy-utility trade-off.
>
> [1] The algorithmic foundations of differential privacy, *Foundations and trends in theoretical computer science* (2014)
>
> **Q1. Clarification on utility advantage in Theorem 3**
>
> Adapting Lemma 3, which proves the convergence of optimal weight $\tilde{\mathbf{w}}$ and the solution of gradient update $\mathbf{w}_T$ in a convex scenario, we prove the utility advantage in terms of convergence. We theoretically prove that the reduced noise level in the all-in-one classifier results in smaller error compared to non-private updated points.
> **Since the utility-privacy trade-off indicates how much performance we sacrifice compared to the non-private model**, the smaller convergence gap between private and non-private models demonstrates the utility advantage of our private model with reduced noise addition.
>
> To clarify our argument, we will change line 188 to “Let $\mathbf{w_T}^{ (\tau)}$ be the $T$‑th iteration with noise $\boldsymbol{z}_\tau$ and $\mathbf{w}_T$ is without noise addition”.
>
> **W2. Lack of formal discussion on improved privacy-utility trade-off**
>
> In **Q5**, we explain why our method achieves better privacy-utility trade-offs through better privacy budget allocation in multi-class settings. Similarly, for gradient updates, the reduced noise level improves convergence toward the non-private optimum, as proven in Theorem 3 (line 186), detailed in **Q1**.
>
> Various experimental results support our hypothesis about privacy-utility trade-off, which means that we obtain higher accuracies in the same privacy budget $\epsilon$ or we require a smaller privacy budget $\epsilon$ to get similar performance.
>
> **Q2. Technical contribution of Theorem 1 (and Typo on Line 377).**
>
> Firstly, we argue that Theorem 1 is derived with our own proof, which is not directly obtained from Lemma 1. Our method follows **a similar proof** of Lemma 1, rather than directly utilizing it. When taking a closer look at the proof of Theorem 1 in Appendix B.1, we can summarize the novelty of the proof in Theorem 1 as follows:
>
> 1. We first prove that the bound of $\mathbf{w}$ of all-in-one SVM
> 2. Then, we apply the sensitivity to bound the weight parameters in the all-in-one SVM
> 3. The calculated bound is tighter than o other DP-ERM methods
>
> To push further, our proof is totally different from that of PrivateSVM, which most DP-ERM approaches rely on. We found that extending PrivateSVM's method to the all-in-one SVM is non-trivial, and our proof gives a tighter bound of sensitivity.
>
> After consideration, we agree that the current Lemma 1 can be confusing to the readers about our technical contribution, so we will remove Lemma 1 (originally from [1]) and replace it with a new lemma that we have proved. The lemma specific to our all-in-one SVM can be stated as follows:
>
> ---
>
> For a convex function $T$, a dataset $D$, and input scaler $g(\cdot)$, let $\tilde{\mathbf{w}} _D = \sum _{i=1}^n \tilde{\alpha}_i g(\mathbf{x}_i)$, where $(\tilde{\alpha}_1,\ldots,\tilde{\alpha}_n)$ is the solution to:
>
> $$\min _{\boldsymbol{\alpha}} \left( \frac{1}{2} \sum _{i,p} \sum _{j,q} \sum _{k} \alpha _{i,p} \alpha _{j,q} \nu _{y_i,p} \nu _{y_j,q} g(\mathbf{x}_i)^T g(\mathbf{x}_j) + \sum _{i,p} T(-\alpha _{i,p}) \right)$$
>
> Let $D^n$ be $D$ with the $n$-th point $\mathbf{x}_n$ removed, and let $\tilde{\mathbf{w}} _{D^n}$ be defined similarly. Then the difference of the weights between original and leave-one-out SVMs is bounded as:
> $\sum _{k=1}^c ||\mathbf{w}_k^{[n]} - \mathbf{w}_k||^2 \leq \lambda _{\max}(G) \|\tilde{\alpha} _{n}\|^2 ||g(\mathbf{x}_n)||^2$.
>
> ---
>
> **Typo.** Thus, for Line 377, the function $T$ in our proof does not need to be the same as that in Lemma 1. If we were to strictly follow the notation of the current Lemma 1, the function $T$ should be written as $T(-\alpha_i, \textbf{x}_i) = -\alpha_i$. However, since $T$ is a constant function with respect to $\textbf{x}_i$, we denoted $T$ as $T(-\alpha _{i,p}) = -\alpha _{i,p}$ for simplify.
>
> [1] Leave-one-out bounds for kernel methods. *Neural computation (2003).*
>
> **We will continue the response in the comments below.**

---

> > ### Comment · Reviewer_N89T · 2025-08-03
> > **Thank for responding to my concerns**
> >
> > I appreciate the diligent response from the authors. Below, I would like to summarize a few key points from authors' rebuttal.
> >
> > 1. >In a non-private scenario, it is hard to say that all-in-one methods are significantly preferred over OvR classifiers in practice. However, in the privacy domain, we can dramatically reduce the privacy budget of multi-class classification to the same level as binary classification.
> >
> > I agree that this is indeed a good contribution. Thanks for your detailed explanation on this point.
> >
> >
> > 2. >Thanks to the reviewer’s comment, we will change Remark 1 into the above formulation and add a detailed explanation about the precise relationship between per-sample access frequency and the privacy-utility trade-off.
> >
> > This would indeed strength the discourse of the paper. Thank you.
> >
> > 3. >Various experimental results support our hypothesis about privacy-utility trade-off, ...
> >
> > I recognize the value of the experiments provided in the paper. I also appreciate authors' response to my concern on theoretical analysis on improved privacy-utility trade-off.
> >
> > 4. >After consideration, we agree that the current Lemma 1 can be confusing to the readers about our technical contribution,...
> >
> > Thank you for providing a detailed explanation on the differences in the proof from prior work.
> >
> > 5. >However, since $T$ is a constant function with respect to $x$ , we denoted ...
> >
> > For some reason, I could not find this explanation in the paper. The fact that you are using $T$ for both function and the number of total steps is confusing. I would recommend using a different notation for one of them.

---

> > > ### Author Response · Authors · 2025-08-04
> > > **Thank for detailed comments**
> > >
> > > We sincerely appreciate the reviewer’s kind and detailed comments on our rebuttal.
> > >
> > > As suggested, we will further clarify the explanation of $T$ and verify it when we address the proofs.
> > >
> > > Thanks to the reviewer’s detailed question, we are delighted that the quality of our work has improved.

---

> ### Author Response · Authors · 2025-08-01
>
> **Q3**. **Novelty of Theorem 2**
>
> We agree that Theorem 2 relies on the moments accountants and DP‑SGD. However, the paper presents several novel approaches for DP‑SVMs, as discussed in **W1/W3** and **Q2**. For gradient perturbation, we focus instead on the compatibility of the DP gradient update.
>
> The main reason to use clipping‑based gradient updates in DP‑SGD is that these approaches are extensively studied, allowing us to leverage recent analytical techniques for gradient‑based optimization. As noted on line 228, we use Opacus, a modern library developed by Meta Open Source. Opacus is actively updated with new DP‑SGD algorithms, including optimizers and sampling strategies. For the adaptive gradient perturbation in Equation 16, we employ the DP‑Adam [1] optimizer, which is officially supported by the Opacus library. Thus, adopting advanced techniques is also feasible within our framework. Other reviewers [hEqw, orwF] also encouraged using recent DP‑SGD techniques to improve performance. We kindly request the reviewer to take a look the rebuttal of **W1, Q1 Sub-sampling techniques** in reviewer [hEqw] how the recent sub-sampling for DP-SGD experimentally improve the performance.
>
> Compared to loss functions commonly used in deep learning, the convex SVM loss yields better convergence, as shown in Theorem 3, even with full‑batch gradient updates. While alternative optimization procedures tailored to SVM optimization are possible, this is a substantial topic. It is beyond the scope of this paper, and we leave it for future work.
>
> [1] Large Language Models Can Be Strong Differentially Private Learners, ICLR 2022
>
> **Q4, Q6. Notation**
>
> **Eq 7.** As the reviewer suggested, we will revise the notation of Equation (7) as follows:
>
> \begin{cases}
> f_{kl}(\mathbf x_i) \ge 1 - \xi_{ikl}, & \text{where } y_i = k \\\\
> f_{kl}(\mathbf x_i) \le -1 + \xi_{ikl}, & \text{where } y_i = l
> \end{cases}
>
> **Eq 16.** As we follow the DP-Adam [1] update, we will update the equation precisely.
>
> When we consider the private gradient of Theorem 2 as $\hat {\mathbf g}_ t$, the adaptive update formulation is as follows:
>
> $$\hat{\mathbf m} _ t = \beta_1 \hat{\mathbf m} _ {t-1} + (1-\beta_1)\hat{\mathbf g} _ t,\ \hat{\mathbf v} _ t = \beta_2 \hat{\mathbf v} _ {t-1} + (1-\beta_2)(\hat{\mathbf g} _ t \odot \hat{\mathbf g} _ t)$$
>
> $$\hat{\mathbf m} _ t = \frac{\hat{\mathbf m} _ t}{1-\beta_1^t},\ \hat{\mathbf v} _ t = \frac{\hat{\mathbf v} _ t}{1-\beta_2^t},\ \hat  {\mathbf w} _ t = \hat {\mathbf w} _ {t-1} - \eta\frac{\hat{\mathbf m} _ t}{\sqrt{\hat{\mathbf v} _ t} + \gamma}.$$
>
> [1] Large Language Models Can Be Strong Differentially Private Learners, ICLR 2022
>
> **[General]** Additionally, we kindly request the reviewer to take a look of any interesting topics during the rebuttal period. In summary, we received positive feedback on: (i) reducing redundant privacy budget consumption for multi-class SVM in DP scenarios [Fj1B, orwF, hEqw, EHLs], (ii) empirical strengths [N89T, Fj1B, hEqw, EHLs], and (iii) well-written and well-executed paper [Fj1B, hEqw].
>
> For each rebuttal concerning “weakness” and “question”, we addressed the following topics:
>
> - [**N89T**] Our novelty and the importance of single data access in DP, and details of theorems
> - [**Fj1B**] Clarity of Lemma 1, convexity analysis of Eq. (12), and selection of hyperparameter $R$
> - [**orwF**] DPSVM in LLM era, selection of all-in-one base model, and advanced techniques in DP-SGD
> - [**hEqw**] Sub-sampling in gradient descent, low performance, and limitations
> - [**EHLs**] Inefficiency, additional comparison methods/related works, membership inference attack, and realistic impacts
>
> If the reviewer has any questions about these topics, please refer to our rebuttals for the corresponding reviewers.

---

> > ### Comment · Reviewer_N89T · 2025-08-03
> > **Thank you for your response to my concerns (Part 2)**
> >
> > I thank the authors for taking note of my concerns, and for responding to them in a detailed manner.
> > Taking into account the authors' rebuttal, and other reviewers' comments, I am happy to increase my score to 4.

---

### Note · Authors · 2025-08-12

Dear Program Chairs, Senior Area Chairs, Area Chairs, and Reviewers,

We would like to express our sincere gratitude to all reviewers for their thoughtful feedback during the rebuttal period. Through constructive conversations, we addressed their concerns across theoretical and experimental dimensions.

Fortunately, all of the reviewers recognized the value of our responses and subsequently raised their scores or maintained their positive scores.

The rebuttal can be summarized as follows:

- We clarified the presentation of mathematical proofs and revised the missing parts
- We provided additional empirical support for our theoretical contributions through experiments with varying batch sizes  and clipping thresholds for DP, as well as evaluation against membership inference attacks
- We expanded the related work section and discussed the strength of SVM in the LLM era

We believe that we have taken all reviewer comments seriously to the best of our ability, and the revisions have significantly improved the quality of our paper.

We would like to express our sincere appreciation to the Reviewers, Program Chairs, Senior Area Chairs, and Area Chairs for their guidance and commitment to the review process. We look forward to your final decision and hope our contribution aligns well with the conference's mission.

Best regards,

Authors

---

### Decision · Program_Chairs · 2025-09-17

**Decision:**

Accept (poster)

**Comment:**

While past work on private SVMs has focused on binary classification, this paper tackles multiclass classification. It argues that methods based on reducing the problem to many binary classifiers do not give a good privacy-utility trade-off as the number of classes grows. Instead, the authors privatize an "all-in-one" multiclass SVM formulation that requires only a single access to each data sample, thereby reducing the privacy composition cost. The paper addresses a classic problem, and its empirical studies show that this approach often outperforms previous methods. All reviewers support the paper.